# Transcriptomic profiling of Schlemm's canal cells reveals a lymphatic-biased identity and three major cell states

Revathi Balasubramanian[1†], Krishnakumar Kizhatil[2†], Taibo Li[3], Nicholas Tolman[1,4], Aakriti Bhandari[1,5], Graham Clark[2], Violet Bupp-Chickering[1], Ruth A Kelly[6], Sally Zhou[1,7], John Peregrin[1], Marina Simón[1], Christa Montgomery[1], W Daniel Stamer[6], Jiang Qian[8], Simon WM John[9]*

[1]Department of Ophthalmology, Columbia University Irving Medical Center, New York, United States; [2]Department of Ophthalmology and Visual Sciences, The Ohio State University Medical Center, Columbus, United States; [3]Department of Molecular Biology and Genetics, Johns Hopkins University, Baltimore, United States; [4]Graduate School of Biomedical Sciences, Tufts University School of Medicine, Boston, United States; [5]Neuroscience Graduate Program, University of Utah, Salt Lake City, United States; [6]Department of Ophthalmology, Duke University, Durham, United States; [7]SUNY Downstate Health Sciences University, New York, United States; [8]Department of Ophthalmology, Johns Hopkins School of Medicine, Baltimore, United States; [9]Zuckerman Mind Brain Behavior Institute, Columbia University, New York, United States

*For correspondence:
sj2967@cumc.columbia.edu

†These authors contributed equally to this work

Competing interest: The authors declare that no competing interests exist.

**Abstract** Schlemm's canal (SC) is central in intraocular pressure regulation but requires much characterization. It has distinct inner and outer walls, each composed of Schlemm's canal endothelial cells (SECs) with different morphologies and functions. Recent transcriptomic studies of the anterior segment added important knowledge, but were limited in power by SEC numbers or did not focus on SC. To gain a more comprehensive understanding of SC biology, we performed bulk RNA sequencing on C57BL/6 J SC, blood vessel, and lymphatic endothelial cells from limbal tissue (~4,500 SECs). We also analyzed mouse limbal tissues by single-cell and single-nucleus RNA sequencing (C57BL/6 J and 129/Sj strains), successfully sequencing 903 individual SECs. Together, these datasets confirm that SC has molecular characteristics of both blood and lymphatic endothelia with a lymphatic phenotype predominating. SECs are enriched in pathways that regulate cell-cell junction formation pointing to the importance of junctions in determining SC fluid permeability. Importantly, and for the first time, our analyses characterize three molecular classes of SECs, molecularly distinguishing inner wall from outer wall SECs and discovering two inner wall cell states that likely result from local environmental differences. Further, and based on ligand and receptor expression patterns, we document key interactions between SECs and cells of the adjacent trabecular meshwork (TM) drainage tissue. Also, we present cell type expression for a collection of human glaucoma genes. These data provide a new molecular foundation that will enable the functional dissection of key homeostatic processes mediated by SECs as well as the development of new glaucoma therapeutics.

## eLife assessment

This **valuable** study has characterized the unique expression of Schlemm's canal endothelial cells (SECs) using FACS-sorted specific cell bulk RNA-Seq and scRNA-/snRNA-Seq of mouse SECs. The

**compelling** study identified novel biomarkers for SECs and molecular markers for two inner wall SEC states and outwall SECs in mouse eyes. Significant gene networks and pathways were elucidated for their potential contribution to glaucoma pathogenesis, providing targets for further research in relation to glaucoma.

## Introduction

Glaucoma is a leading cause of blindness affecting 80 million people (*Tham et al., 2014*). Elevation of intraocular pressure (IOP) is a major risk factor for glaucoma. Current glaucoma treatments reduce IOP using drugs and surgeries that decrease aqueous humor (AQH) formation or increase aqueous humor drainage (outflow) from the eye (*Weinreb et al., 2014*). Abnormally increased outflow resistance results in elevation of IOP and contributes to glaucoma. Schlemm's canal (SC) and the trabecular meshwork (TM), key tissues of the conventional outflow pathway, are critical in regulating IOP and ocular fluid homeostasis. The flow resistance of the conventional outflow pathway is an important determinant of IOP. SC is an endothelial vessel circumscribing the eye within the iridocorneal angle at the limbus, where the iris and cornea meet. The canal has inner and outer walls composed of SECs (*Lewczuk et al., 2022*). The inner wall (IW) of SC is the final barrier that AQH must cross before it passes into the lumen of SC, which is connected to venous circulation by vascular branches known as collector channels. The IW is specialized for its drainage functions with higher expression of lymphatic genes and differing cellular morphology compared to the outer wall (OW; *Kizhatil et al., 2014*). The IW regulates aqueous humor drainage into the canal's lumen. IW cells respond biomechanically to regulate IW fluid permeability and drainage. Specifically, they respond to changes in ocular pressure and shear stress, generated by AQH flow, forming specialized drainage structures called giant vacuoles as well as pores through which AQH passes (*Braakman et al., 2016*; *Vahabikashi et al., 2019*; *Braakman et al., 2015*). However, much remains to be discovered about the mechanisms that determine outflow resistance and mediate outflow under both normal and pathological conditions.

Delineating the molecular control of outflow resistance and of inner wall fluid permeability is important for understanding IOP homeostasis and developing novel glaucoma treatments. A key site of resistance to outflow is located where SC and the adjacent juxtacanalicular meshwork (JCT) – a subpart of the TM – meet. The SC and TM are intimately linked at the cellular, biomechanical, and functional levels (*Stamer et al., 2015*; *Overby et al., 2014*). Both physical and molecular interactions between SC and TM cells participate in tissue formation, tissue maintenance and the regulation of resistance to outflow. The mechanisms by which the SC and TM coordinately regulate resistance to outflow and IOP are still being elucidated.

SC is a highly specialized vessel with its endothelial cells having properties of both blood vessels and lymphatic endothelial cells (BECs and LECs) (*Kizhatil et al., 2014*). It develops by a process known as canalogenesis, which progresses through stages with features of angiogenesis, vasculogenesis, and lymphangiogenesis (*Kizhatil et al., 2014*; *Park et al., 2014*; *Aspelund et al., 2014*; *Karpinich and Caron, 2014*). Analyses of the genes linked to glaucoma most strongly identify lymphangiogenesis and endothelial processes, strongly implicating dysfunction of SC (*Hamel et al., 2022*). Despite this, SC is less studied than the TM and so we focus on SC in this study.

Single-cell RNA sequencing (scRNA-seq) is revealing the cellular heterogeneity and transcriptomic profiles of various ocular cell types, including the anterior segment from several species (*van Zyl et al., 2020*; *van Zyl et al., 2022*; *Monavarfeshani et al., 2023*; *Patel et al., 2020*; *Thomson et al., 2021*). These important studies included SECs, but were underpowered and limited for SC due to the low representation of SECs in the studied tissues and the very small numbers of SECs successfully sequenced. To more deeply and rigorously characterize SECs, we here provide a multimodal transcriptomic analysis. We present bulk sequencing data for SECs, LECs, and BECs from the same tissue sample, providing averaged data on SEC gene expression at depth for ~4500 SECs. We compare these bulk data to single cell (sc) and single nucleus (sn) RNA sequencing and then use the individual SEC transcriptome data to characterize SEC heterogeneity (across transcriptomes from 903 cells/nuclei from two strains of mice; C57BL/6 J scRNA-seq 166, C57BL/6 J snRNA-seq 375, 129/Sj scRNA-seq 362). Both the bulk and single cell resolution data confirm that SEC transcriptomes have similarities to BECs and LECs, but with an important bias towards lymphatic similarity. By sequencing a larger number of limbal SECs than previously reported, our study adds significant power

and robustness, uncovering important new molecular characteristics of SECs. Building on the previous studies, it provides a more solid foundation to guide future work. Importantly, we identify unique gene expression features that distinguish inner and outer wall SECs, and we validate expression patterns using immunofluorescence (IF). Using pathway analysis and predicted ligand-receptor analysis, we provide critical new information about genes and pathways functioning in SC and about molecular interactions between SC and TM that are predicted to be central for tissue homeostasis, IOP control, and glaucoma. These unbiased methods greatly expand our molecular knowledge of SC.

## Results

### Schlemm's canal ECs are more similar to lymphatic than blood vessel ECs

To provide a more detailed understanding of the molecular nature of SECs, we compared the transcriptome of isolated limbal endothelial cells (SECs, BECs and LECs) obtained by bulk RNA sequencing. We developed a cell sorting method to isolate cells while preserving SECs. Limbal strips from Prox1-GFP mice were gently dissociated and dispersed cells were prelabeled with antibodies against LYVE1 and endomucin (EMCN). FACS was used to separate the dispersed cells into SECs (GFP+, EMCN+), LECs (GFP+, LYVE1+, EMCN-), and BECs (EMCN +only). When we isolated mouse SECs with standard tissue dissociation procedures very few cells were obtained (<<100 from 10 eyes), whereas our optimized protocol allowed isolation of sufficient cells for meaningful studies (1500–1700 SECs, 200–300 LECs, and 2000–2500 BECs from 10 eyes) (*Figure 1—figure supplement 1*).

Pair-wise comparisons of the bulk RNA-seq data for each purified cell type demonstrate that the transcriptome of SECs displays more genes in common with LECs than BECs (*Figure 1*). As LECs and SECs have the most similar transcriptomes, volcano plots of differentially expressed genes (DEGs) between LECs and SECs showed fewer DEGs than for BECs versus SECs or for LECs versus BECs (*Figure 1A*). *Npnt* was highly expressed in SECs, but absent in LECs and BECs. As expected (*Kizhatil et al., 2014*; *Park et al., 2014*), *Prox1* was the top DEG comparing SECs versus BECs and also LECs versus BECs (*Figure 1A*). The dendrogram distances after hierarchical clustering and the signature gene heat maps of the bulk RNA sequencing data further demonstrate that the transcriptome of SECs is closer to that of LECs than BECs (*Figure 1B*).

We next determined which canonical pathways were different between the cells based on their transcriptomes. Gene set enrichment analysis (GSEA) of SECs compared to BECs or LECs showed a common theme with extracellular matrix interactions (ECM glycoprotein, integrin cell surface interactions) being highly enriched in SECs compared to both other cell types (*Figure 1C*). TGFβ (*Nakamura et al., 2021*; *Li et al., 2022*) cell signaling pathway was also enriched in SECs when compared to BECs but not LECs. Additionally, interleukin pathways were enriched in SECs compared to both BECs and LECs indicating their specific potential to interact with immune cells (*Figure 1C*, *Supplementary file 1*, *Supplementary file 2*).

### Cell types in mouse limbal tissue

Using scRNA-seq, we successfully sequenced 9272 single cells from dissected limbal tissue (enriched for drainage structures) of C57BL/6 J mice (*Figure 1—figure supplement 1*). An unbiased low-resolution clustering of single cells projected onto a Uniform Manifold Approximation and Projection for Dimension Reduction (UMAP) space identified 9 major clusters representing various anterior segment cell types (*Figure 2A*). Clusters were assigned to their cell types using previously characterized marker genes (*van Zyl et al., 2020*) as indicated in the figures (*Figure 2—figure supplement 1*). We then focused on the *Egfl7*-expressing EC (a commonly used endothelial marker, 469 cells) cluster for further sub-clustering and analyses (*Figure 2B*).

### SEC transcriptome is unique but closer to LEC than BEC

Using the strain C57BL/6 J scRNA-seq data, the *Egfl7*-expressing ECs subclustered (unbiased methods) into three distinct sub-clusters (*Figure 2C and D*). Based on well-characterized marker gene expression (*Kizhatil et al., 2014*; *Park et al., 2014*; *van Zyl et al., 2020*; *Kalucka et al., 2020*) these sub-clusters were assigned as BEC (281 cells, high level of *Flt1 /VegfR1*), LEC (22 cells, high level of *Prox1* and *Lyve1*), or SEC (166 cells expressing *Prox1* but not *Lyve1*; *Figure 2D*). Among ECs, *Npnt*

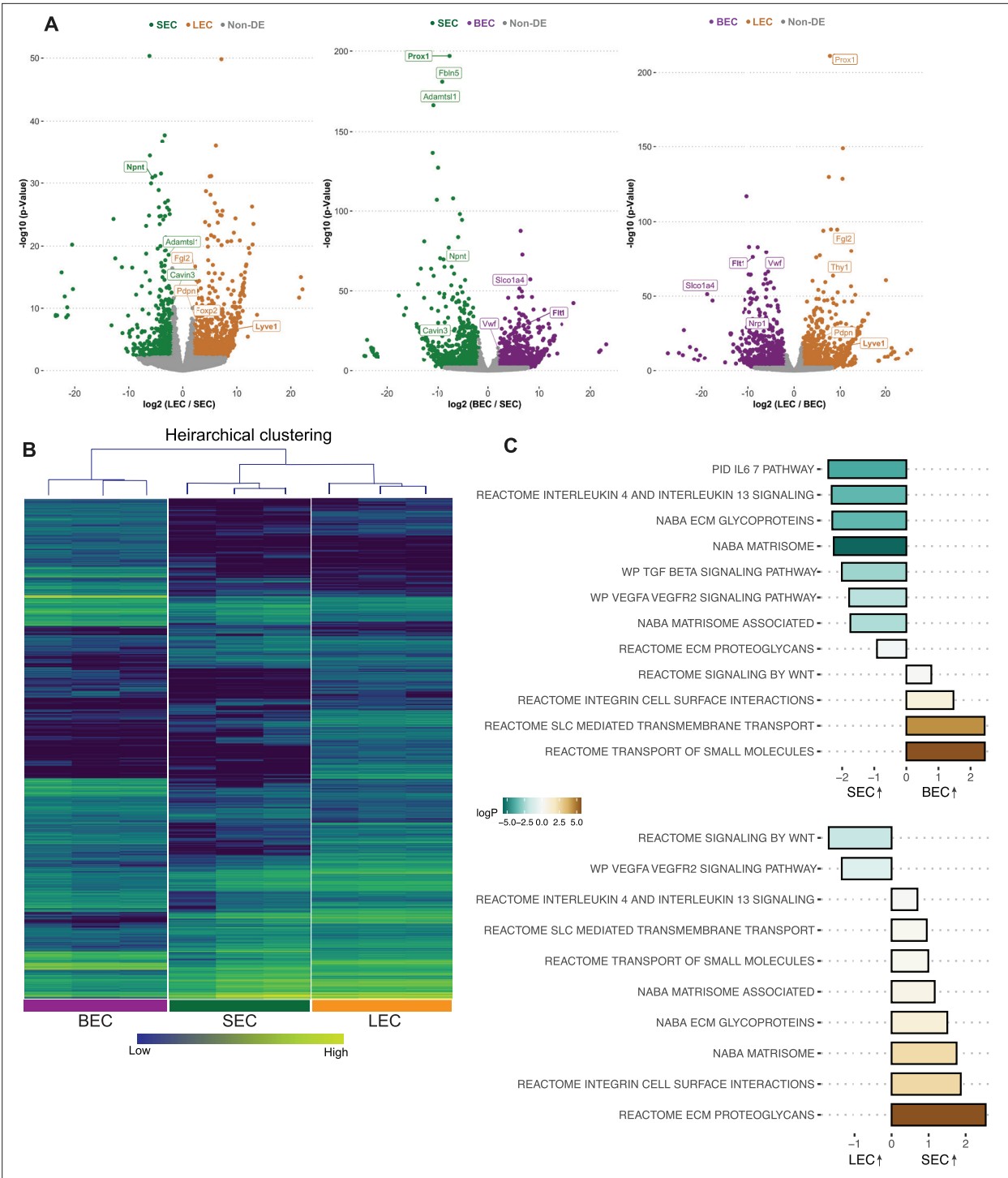

**Figure 1.** SEC bulk transcriptome is more similar to LEC than BEC transcriptomes. (**A**) Pairwise comparison of bulk RNA sequencing data showing differentially expressed genes (DEGs) between cell groups, note larger number of non-DEGs in gray in plot of LEC vs. SEC compared to others. (**B**) Hierarchical clustering of BECs, SECs, and LECs. (**C**) Pathways enriched in SECs vs BECs (top panel), LECs vs. SECs (bottom panel) by GSEA analysis. BEC: Blood endothelial cell, LEC: Lymphatic endothelial cell, SEC: Schlemm's canal endothelial cell.

The online version of this article includes the following figure supplement(s) for figure 1:

**Figure supplement 1.** RNA sequencing of anterior segment limbal tissue.

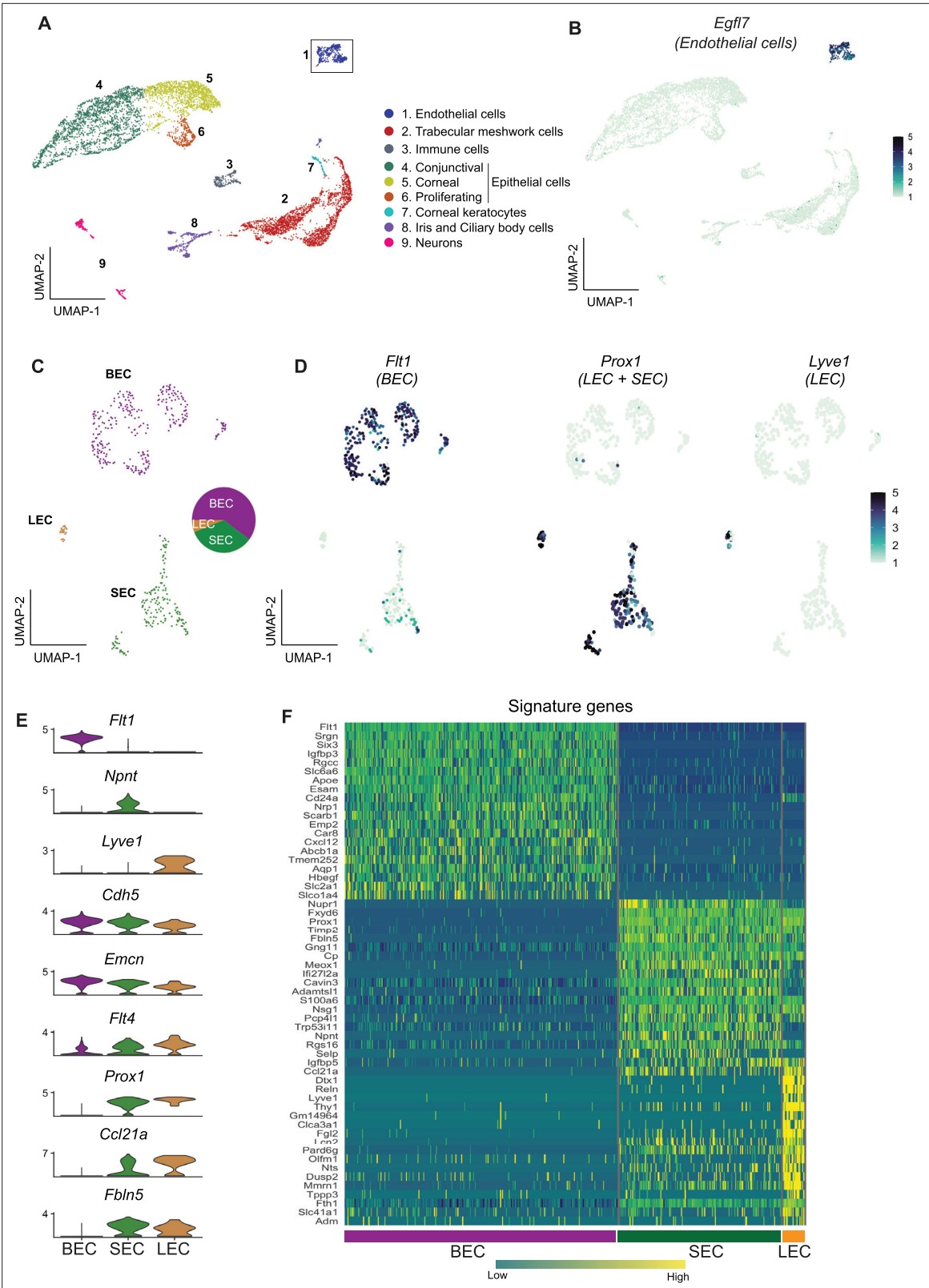

**Figure 2.** scRNA-seq data reveals robust SEC signature genes. (**A**) Clusters of cells from the limbal tissue represented on a UMAP space (**B**) Expression of *Egfl7* in endothelial cells. (**C**) Endothelial cell cluster is sub-clustered into BEC, LEC, and SEC. (**D**) Expression of *Flt1* enriched in BEC, *Prox1* in LEC and SEC, and *Lvye1* in LEC identifying the three cell clusters. (**E**) Violin plot of genes expressed BEC, SEC, and LEC from scRNA-seq data. (**F**) Heatmap

*Figure 2 continued on next page*

*Figure 2 continued*

of signature genes from BEC, SEC, and LEC from scRNA-seq data. BEC: Blood endothelial cell, LEC: Lymphatic endothelial cell, SEC: Schlemm's canal endothelial cell.

The online version of this article includes the following figure supplement(s) for figure 2:

**Figure supplement 1.** Cell types in mouse limbal tissue.

**Figure supplement 2.** Correlation between sequencing modalities.

is uniquely expressed in SECs, *Cdh5,* and *Emcn* are expressed at high levels in BECs, but only at mid and low levels in SECs and LECs, respectively. *Flt4* (*VegfR3*) is expressed at high levels in LECs, and at mid or low levels in SECs and BECs, respectively. *Prox1*, *Ccl21a*, and *Fbln5* are expressed in both LECs and SECs. *Prox1* and *Ccl21a* are expressed at higher levels in LECs but are not unique to either LECs or SECs (*Figure 2E*). A heatmap of the highly represented genes for each cell type indicates that the transcriptome of SECs more closely aligns with that of LECs than BECs (*Figure 2F*). The scRNA-seq data correlated well with the bulk sequencing data (*Figure 2—figure supplement 2*). When analyzing scRNA-seq data using Seurat, clusters are determined by calculating shared nearest neighbors followed by modularity optimization using a Louvian algorithm. To determine if the signature genes identified in these clusters identified by Seurat were robust and able to distinguish cell types in bulk RNA-seq samples, we used this signature gene list for unsupervised hierarchical clustering of the bulk RNA-seq data. This correctly identified the three cell types producing accurate SEC, BEC, and LEC clusters. This demonstrates the robustness of the signature genes across independent samples using distinct statistical calculations (*Figure 2—figure supplement 2*).

## RNAseq enables detection of OW and IW transcriptomes

Previous studies reported higher representation of ECs when using snRNA-seq compared to scRNA-seq (*Wu et al., 2019*; *Wen et al., 2022*) (and personal communication, Seth Blackshaw). Thus, we reasoned that snRNA-seq would capture more ECs and provide more power to discriminate SEC subtypes using unbiased methods. Using snRNA-seq, we analyzed 10,764 C57BL/6 J nuclei (*Figure 3A*) and captured 1287 EC nuclei (11.9% of total nuclei). This is a clear enrichment over scRNA-seq (EC 5.1% of cells).

The scRNA-seq and snRNA-seq datasets integrated well and agreed with each other in all assessed ways (*Figure 3B*). Although snRNA-seq captured fewer distinct genes (median gene count per cell = 759) than scRNA-seq (median gene count per cell = 2603), endothelial cells from the two datasets had a high degree of correlation of gene expression (*Figure 2—figure supplement 2*). Integration of *Egfl7* +*Cdh5*+endothelial cells from the two datasets provided transcriptome data for 541 C57BL/6 J SECs (166 scRNA-seq +375 snRNA-seq).

Unbiased clustering of this larger number of SEC transcriptomes separated SECs into two distinct sub-clusters (*Figure 3C and D*). One cluster had higher expression of *Npnt* (representing inner wall, IW, cells; validation in results below) while the other had higher levels of *Selp* and *Sele* (representing outer wall, OW, cells). A small number of the OW cluster cells also had a limited, but specific, expression of *Ackr1* (alias *Darc*), which was previously suggested (*van Zyl et al., 2020*; *Ujiie et al., 2023*) as a marker for collector channels endothelial cells. We manually separated these cells from the outer wall cluster and here forward denote them as collector channel (CC) cells. *Ackr1 is* validated to represent CC's (see Results below). Given the deeper sampling of transcripts of distinct genes in the scRNA-seq data, we manually separated our scRNA-seq SECs into IW, OW, and CC clusters using markers derived from the unbiased clustering of integrated data (*Figure 3—figure supplement 1*).

## Additional scRNA-seq transcriptomes allow detection of two distinct IW cell states

We reasoned that adding more transcriptomes with deeper coverage (transcripts from greater number of distinct genes detected) would enable better resolution of SEC subclasses. To examine more transcriptomes and with deeper coverage than snRNA-seq can achieve, and to validate our findings in a genetically divergent strain of mice, we performed scRNA-seq on limbal tissue from 129/Sj mice. We successfully sequenced 8812 limbal cells of 129/Sj mice (median gene count per cell = 1144). All major cell types were represented in the 129/Sj limbal data (*Figure 4A*) and it compared well to the C57BL/6 J data. We integrated the *Egfl7* +*Cdh5*+endothelial cells from 129/Sj and C57BL/6 J

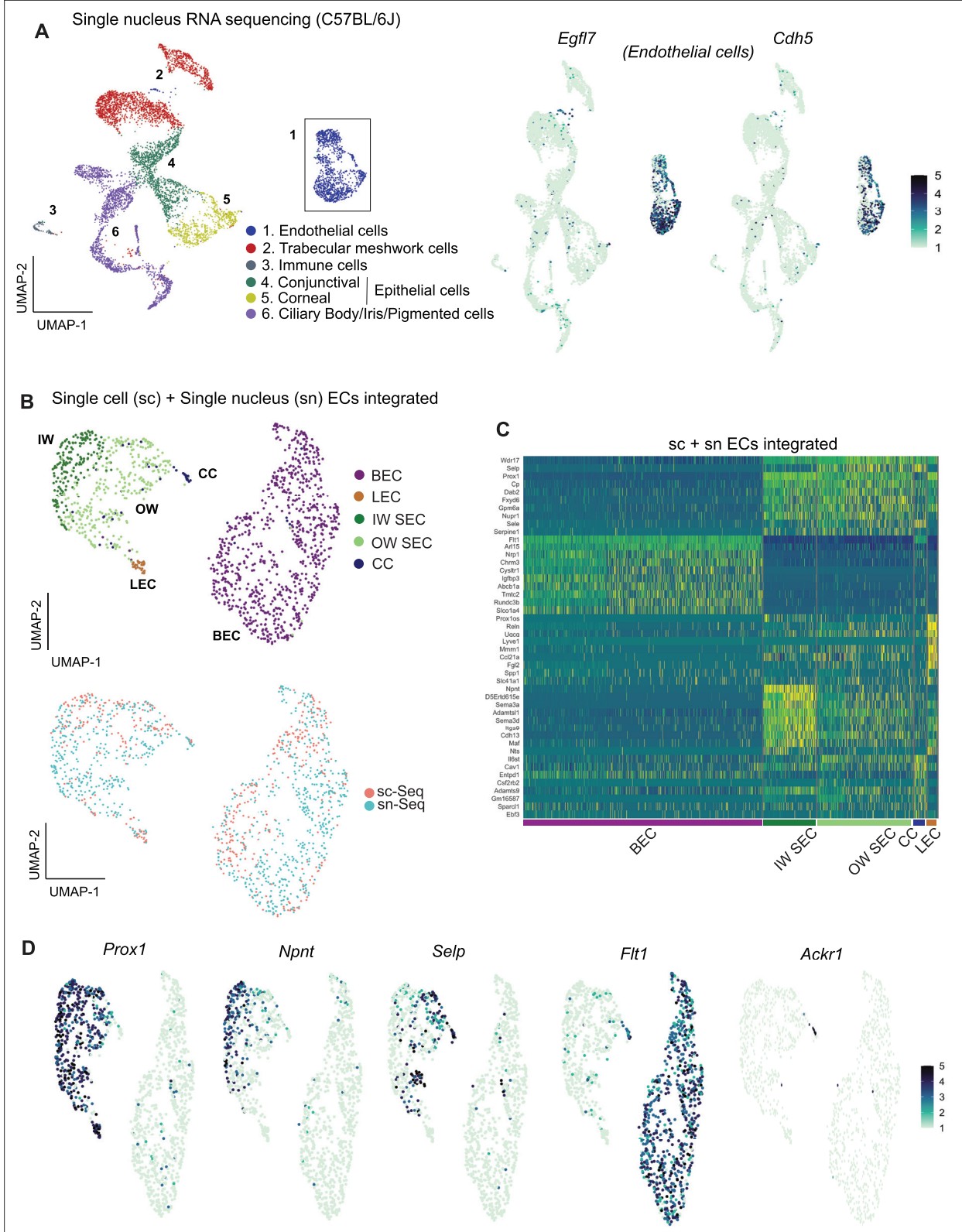

**Figure 3.** Integrating scRNA and snRNA-seq enables detection of outer and inner wall transcriptomes. (**A**) snRNA-seq of C57BL/6 J limbal tissue identifies similar cell types as the sc RNA-seq but captures more endothelial cells (left panel). Expression of *Egfl7* and *Cdh5* in snRNA-seq endothelial cells (right panel). (**B**) Integration of sc and snRNA-seq endothelial cells followed by sub-clustering identifies BECs, LECs, IW SECs, OW SECs, and CC. Integration of sc- and sn- RNA sequencing shows distribution across clusters (bottom panel). (**C**) Heatmap of differentially expressed genes of the

*Figure 3 continued on next page*

*Figure 3 continued*

identified sub-clusters. (**D**) SECs and LEC sub-clusters identified in B, expressing *Prox1*. In the SEC cluster, IW cells express *Npnt and OW* cells *Selp*, CCs express *Ackr1* and BECs robustly express *Flt1*. IW: Inner wall, OW: Outer wall, CC: Collector channels, BEC: Blood endothelial cell, LEC: Lymphatic endothelial cell, SEC: Schlemm's canal endothelial cell.

The online version of this article includes the following figure supplement(s) for figure 3:

**Figure supplement 1.** EC and pericyte transcriptomes in scRNA-seq data guided by integrated multimodal data.

datasets, a total of 1492 ECs (528 SECs; *Figure 4B*). No statistically significant pathway differences were detected between the SECs of these strains, and a full analysis of the effects of strain background is beyond the scope of the current study (*Supplementary file 3*, shows a DEG list). Unbiased clustering of this integrated data detected three SEC states by further resolving the previously detected IW state into two subclasses that we call IW1 and IW2. IW1 cells are typically *Npnt*^high^ and IW2 cells are often *Ccl21a*^high^ while OW cells are often *Selp*^high^, but combinations of markers provide the best cell type resolution (*Figure 4E*, *Supplementary file 5*). We again identified and manually clustered a small group of *Selp*^high^ cells that were *Ackr1+* CC cells (*Figure 4C–E*).

## Immunofluorescence assays validate transcriptomic resolution of OW and IW SECs

To determine the location of cells with detectable NPNT, CCL21A, and SELP proteins in eyes we performed IF on ocular sections and flat mounts of the anterior segment. NPNT was uniquely expressed at a high level in the extracellular matrix surrounding the IW of SC (*Figure 5A*) but was not detected in the OW of SC. NPNT was often detected to have a gradient of expression within SC, with detection at highest levels in the anterior portion of SC (and sometimes exclusively anterior detection). Using in situ hybridization, we confirm the expression of *Npnt* in IW SECs with an anterior bias (*Figure 5—figure supplement 1*). CCL21A often had the reverse gradient, with the greatest (and sometimes exclusive) detection in the posterior portion of SC (*Figure 5A and B*; *Figure 5—figure supplement 1*). This suggests that the expression of NPNT and CCL21A may be modulated by local environmental differences such as aqueous humor flow rates.

*Ccl21a* has been previously used as an SEC marker in scRNA analyses of mouse data (*van Zyl et al., 2020*). However, our current data and a previous study (*Aspelund et al., 2014*) clearly show that CCL21A is not unique to SC in the limbus being also expressed in limbal lymphatics (*Figure 5B*).

SELP was detected in the outer wall of SC and some, but not all, BECs. This agrees with our transcriptomic data, where *Selp* was expressed in a significant subset of SECs by snRNA-seq, but minimally so in BECs. Whole mount staining of the anterior segment revealed SELP expression in OW SECs and ECs of CCs (*Figure 5C*). Taken together, the subgroup of SECs high in *Selp* expression must be OW SECs and CCs, while the subgroup of SECs with obvious *Npnt* expression are IW SECs.

Immunofluorescence detected ACKR1 in CCs and possibly very few OW cells (only where the CCs and OW meet, *Figure 5D*). We denote these cells as CC cells in this paper. The exact starting location of CCs is hard to discern, as they blend into the OW, and there are very few of these *Ackr1*-expressing cells in the transcriptome data. The best ACKR1 antibody that we have tested is FAB6695 which has been used to detect ACKR1 in endothelial cells (*van Zyl et al., 2020*). However, it binds non-specifically to another protein(s) in macrophages (*Kwon et al., 2022*; *Rot et al., 2022*). At the RNA level, the CC cells also express low levels of the SEC marker *Prox1* and have intermediate to high expression of the vascular markers *Flt1* and *Aqp1* (*Figure 2D.*, *Figure 3D*, *Figure 4E*). Thus, we also assessed FLT1 by IF. Agreeing with both the transcriptomic data and our cell type designations, FLT1 was detected strongly in CCs and BECs, and to a weaker extent in a subset of SECs (*Figure 5—figure supplement 2*). Therefore, manual clustering of *Ackr1 +Selp +* cells in the SEC cluster of both scRNA-seq and snRNA-seq accurately identifies CC cells (*Figures 3B, C and 4A–C*).

## Other cell types within the major EC cluster and confirmation of greater lymphatic polarization of IW

Unless otherwise mentioned, further analyses were performed using the C57BL/6 J scRNA-seq data because it had the deepest transcript coverage. In EC sub-clusters, we identified clustering based on sex-specific differences (*Figure 5—figure supplement 2*). We identified a smaller 'male' cluster and a

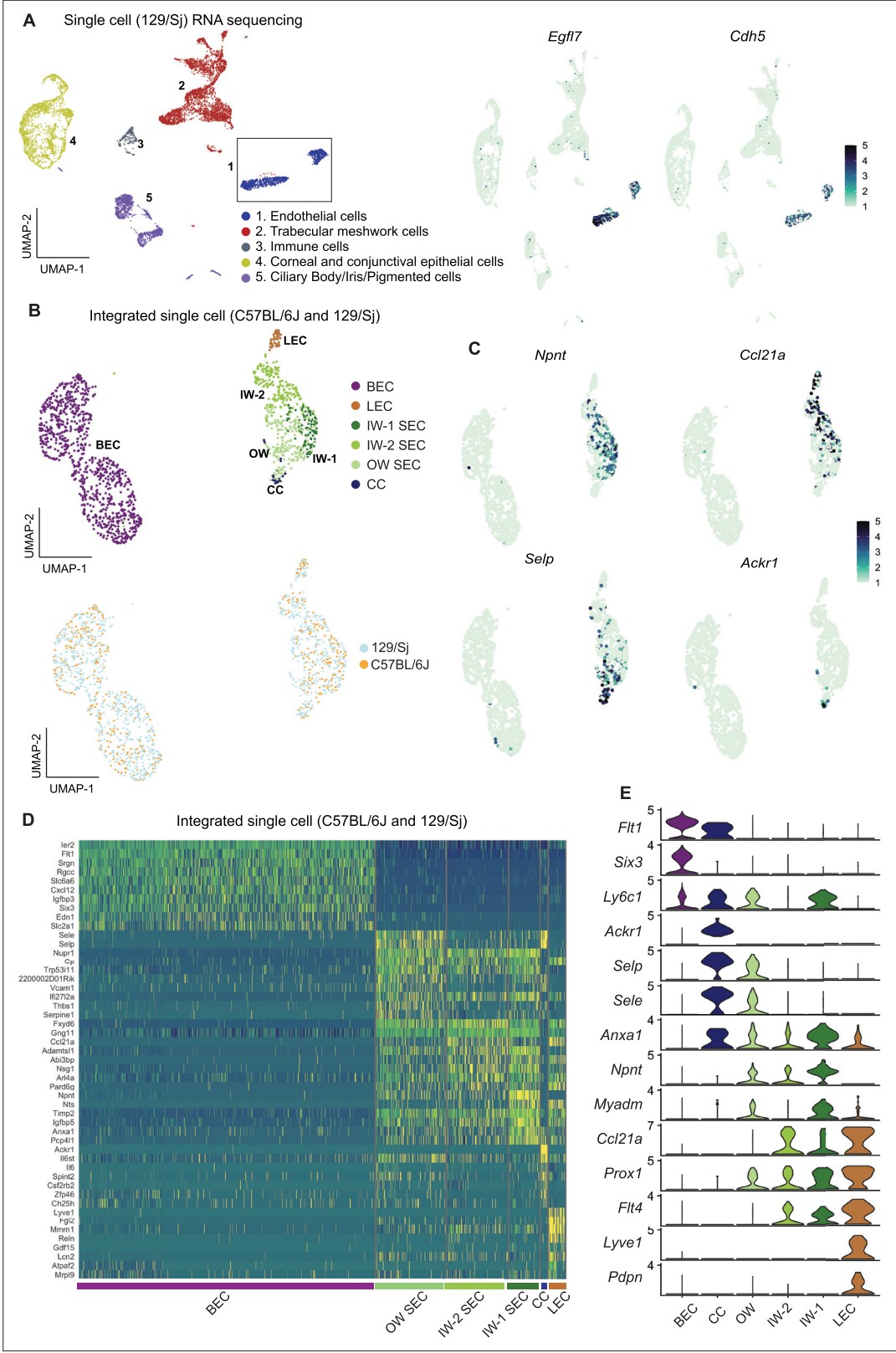

**Figure 4.** Integration of C57BL/6 J and 129/Sj endothelial cells identifies IW states. (**A**) scRNA-seq of 129/Sj anterior segment tissue identifies various cell types similar to that in C57BL/6 J single-cell RNA sequencing (left panel). Expression of *Egfl7* and *Cdh5* in snRNA-seq endothelial cells (right panel). (**B**) Integration of B6 and 129/Sj endothelial cells followed by sub-clustering identifies BECs, LECs, IW1 SECs, IW2 SECs, OW SECs, and CCs (top

*Figure 4 continued on next page*

*Figure 4 continued*

panel). Integration of B6 and 129/Sj endothelial cells distributed across clusters (bottom panel). (**C**) Sub-clustering identifies complementary expression patterns of *Npnt* and *Ccl21a* in IW1 and IW2 SECs, *Ackr1* in CCs, and *Selp* in OW SECs. (**D**) Heatmap of differentially expressed genes of the identified sub-clusters. (**E**) Violin plot showing differences in expression levels of various genes which as a combination defines individual sub-clusters. IW: Inner wall, OW: Outer wall, CC: Collector channels, BEC: Blood endothelial cell, LEC: Lymphatic endothelial cell, SEC: Schlemm's Canal endothelial cell.

larger 'female' cluster in SECs, BECs, and LECs. The smaller male cluster size reflects our use of a third of the number of female eyes. Genes such as *Xist* (involved in the X-inactivation process) and *Ddx3y* (a Y-chromosome-linked gene) were specifically expressed in the female and male clusters respectively. *Lars2* and *Aes* were two genes associated with female and male clusters respectively, consistent with previous studies in other endothelial cell types (*Paik et al., 2020*). Whether there are functional differences resulting from sex-specific gene expression differences remains to be determined.

A close examination of the BEC cluster in the scRNA-seq dataset revealed a sub-cluster of cells that highly expressed markers such as *Des (Desmin)*, *Pdgfrb*, and *Rgs5*, all known markers for pericytes (*He et al., 2018*; *Zhu et al., 2022*). In addition to wrapping around blood vessels, IF analysis showed that the *Des*-expressing pericytes also associated closely with the CCs (*Figure 5—figure supplement 2*). In addition to pericytes, we identified arteries expressing *Sox17*, capillaries expressing high levels of *Esm1*, and veins expressing *Adamtsl1* and *Car4* in the BEC cluster (*Kalucka et al., 2020*; *Barry et al., 2019*; *Cui et al., 2015*; *Chen et al., 2014*; *Figure 5—figure supplement 3*).

Using IF we show expression of lymphatic regulatory molecules including *Prox1* and *Flt4* in SECs, and specifically an enrichment of lymphatic molecules in the IW of Schlemm's canal further confirming (*Kizhatil et al., 2014*) a greater lymphatic polarization of IW cells (*Figure 5—figure supplement 3*).

## Cell type assignment of high IOP and glaucoma genes

We determined the cell type-specific expression of genes associated with elevated IOP and glaucoma by previous genome-wide association (GWAS) studies (*Khawaja et al., 2018*; *MacGregor et al., 2018*; *Choquet et al., 2018*; *Gao et al., 2018*; *Gharahkhani et al., 2021*; *Shiga et al., 2018*) and of developmental glaucoma genes identified by genetic studies (*Figure 6A–B*, *Figure 6—figure supplement 1*). The majority of genes were expressed in endothelial cells and TM cells. A disease score relevance analysis (scDRS) of GWAS genes in the entire limbal tissue indicated that a significant number of these genes were expressed predominantly in TM, endothelial cells, ciliary body, iris cells, and corneal epithelial cells (*Figure 6C*). Of the genes expressed in endothelial cells, we analyzed specific expression in subtypes and found that most genes were expressed broadly among endothelial cell types, supporting the importance of vascular regulatory functions in IOP and glaucoma biology (*Figure 6D*).

## Predicted receptor-ligand analysis reveals signaling pairs between TM and SC

Predictive algorithms trained on known ligand-receptor interactions (*Browaeys et al., 2020*) support active signaling between TM and SC cells. For instance, secreted ANGPT1 ligand expressed by TM cells is predicted to interact with TEK receptor expressed by SECs. ANGPT1/TEK is known to be a key signaling pathway in the developing and adult SC with GWAS linking it to ocular hypertension and glaucoma (*Gharahkhani et al., 2021*; *Thomson et al., 2017*; *Souma et al., 2016*; *Kim et al., 2017*; *Figure 7A*). We also identified secreted VEGFA, VEGFB, and VEGFC from TM with interaction partners on SC cells (*Kizhatil et al., 2014*; *Aspelund et al., 2014*; *Reina-Torres et al., 2017*). In addition to secreted factors, novel contact-based interactions were also predicted, including contact-based signaling between Junctional adhesion molecule 2 (JAM2) in TM cells and ITGB1 on SC cells. Conversely, we examined predicted interactions between ligands expressed by SECs with TM cells. Overall, fewer predicted interactions were found, arguing for a more important role for signaling between ligands originating from the trabecular meshwork in the development, maintenance, and function of SC than vice versa (*Figure 7B*).

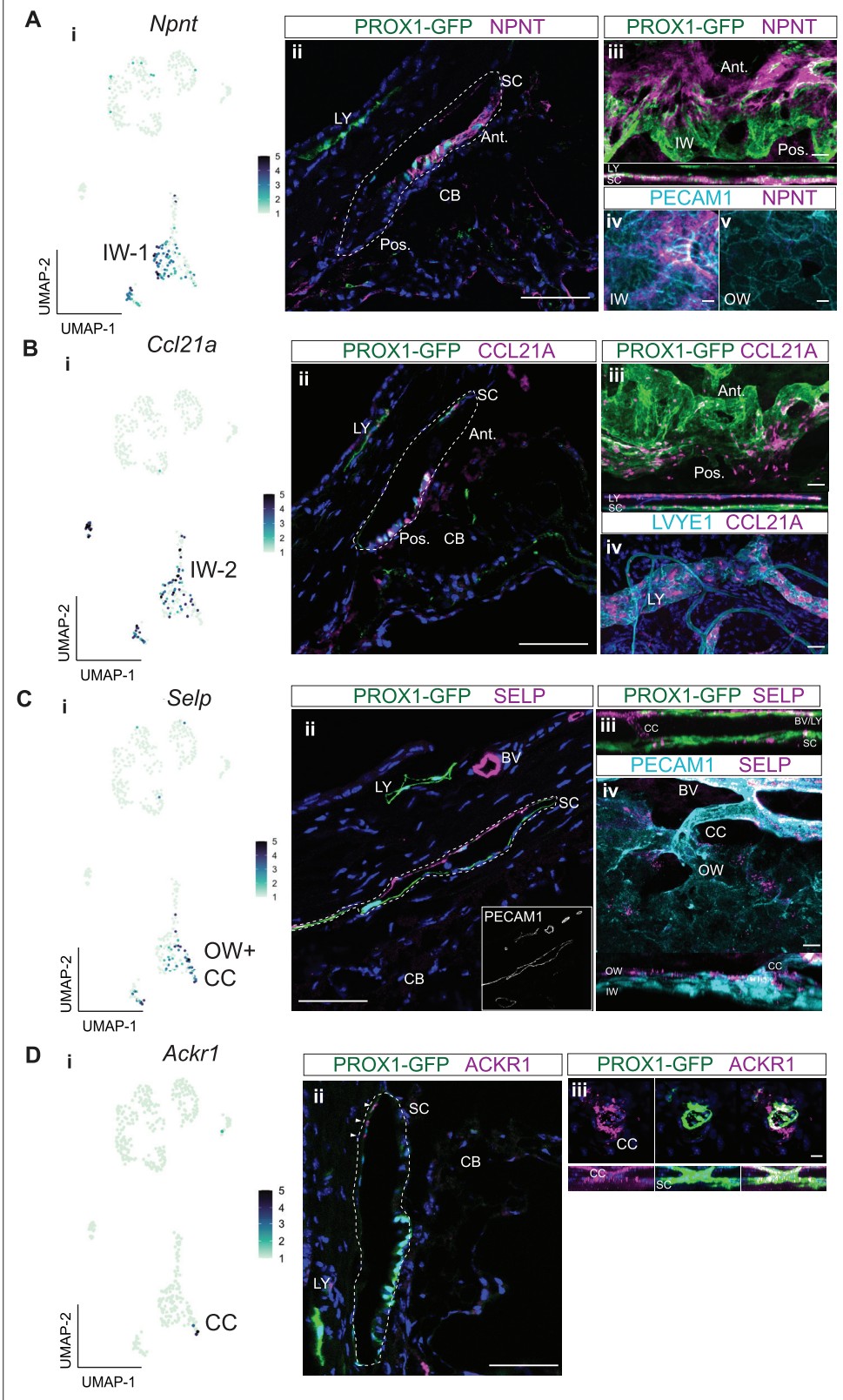

**Figure 5.** Immunofluorescence validates cell types and discovers bias for discrete localization of IW1 and IW2 cells. (**A**) *Npnt* expression in a subgroup of SEC in scRNA-seq data (i) and corresponding immunofluorescence (IF) reveals high level of expression of NPNT in anterior portion of IW of SC in a frozen section (ii) and whole mount (iii and iv). (**B**) *Ccl21a* is expressed in SECs and LECs (i) and corresponding IF reveals high expression in posterior

*Figure 5 continued on next page*

*Figure 5 continued*

portion of IW of SC in a frozen section (ii) and whole mount (iii and iv). (**C**) *Selp* is expressed in OW SECs and CCs, a subgroup of SECs in single-cell (i) and corresponding IF (ii frozen section, iii-iv whole mount). (**D**) *Ackr1* expression in a subset of CC cells (i) and corresponding IF (ii frozen section, iii whole mount). DAPI in blue labels nuclei in all panels. IW: Inner wall, OW: Outer wall, CC: Collector channels, CB: Ciliary body, LY: Lymphatic vessels, BV: Blood vessels SC: Schlemm's canal. Ant.: Anterior SC, Post.: Posterior SC. Scale bar = 100µm.

The online version of this article includes the following figure supplement(s) for figure 5:

**Figure supplement 1.** Biased but variable localization of NPNT and CCL21A in IW of SC.

**Figure supplement 2.** Collector channels, pericytes, and sex-dependent differences within the major EC cluster.

**Figure supplement 3.** Main BEC types within major EC cluster and immunofluorescence confirmation of greater lymphatic polarization of IW.

## Gene ontology enrichment analysis highlights key cellular component/pathways in SECs

Gene ontology (GO) analysis of SC-enriched genes was performed in comparison to a combination of the other endothelial cells and TM cells (*Figure 7C*). Enrichment of the cell-cell junction, membrane raft, and endocytic vesicles cellular components in SECs highlights their importance for SEC maintenance and cellular functions. Molecular pathways enriched in SECs were actin binding, cell adhesion molecule binding, SH3 domain binding, protein tyrosine kinase binding. Enriched biological processes in SECs were actin filament organization and regulation of actin filament-based processes, small GTPase-mediated signaling, cell substrate adhesion, and regulation of vascular development (*Figure 7C*).

These data highlight diverse pathways through which SC is likely to regulate outflow and hence IOP. For example, the nodal, membrane raft pathway includes the caveolin genes, *Cav1* and *Cav2,* which are functionally implicated in intraocular pressure modulation and glaucoma (*Enyong et al., 2022*; *Elliott et al., 2016*) as well as the caveolae-related genes, *Cavin2* and *Cavin3,* with roles in regulating the formation of caveolae (*Kovtun et al., 2015*; *Figure 7D*). Interestingly, *Il6st*, an immune component is also associated with this node and has recently been implicated in regulating the production and function of caveolins (*Schmidt-Arras and Rose-John, 2021*). In the same node are other key vascular regulators that are expressed in SC such as *Kdr* (regulates various functions including permeability), *Tek* (endothelial barrier function), *Plvap* (fenestrae formation), and *Nos3* (vascular homeostasis; *Figure 7—figure supplement 1*). GO analysis of IW or OW enriched genes compared to other endothelial cells was performed on the integrated C57BL/6 J and 129/Sj scRNA-seq data. Among molecular pathways enriched in IW cells are receptor-ligand activity, cell adhesion molecule binding, integrin binding, and calcium-dependent protein binding. Among molecular pathways enriched in OW cells are growth factor binding and glycosaminoglycan binding indicating prominent roles in regulating immune function and chemokine activity in SECs (*Figure 7—figure supplement 2*).

## Discussion
### A multi-modal approach to understanding Schlemm's canal

Single-cell resolution RNA-seq is revolutionizing molecular knowledge of cell types, cell states, and intercellular interactions in health and disease. Both scRNA-seq and snRNA-seq methods are greatly enabling. They overcome the previous necessity to purify individual cell types from complex tissues, and the associated technical issues with doing so. Importantly, they provide previously unavailable information about cell type and cell state heterogeneity. Nevertheless, tissue preparation methods still have to be optimized for specific tissues and cell types. Different preparation methods capture substantially distinct proportions of some cell types and/or different depths of transcriptomic data. Comprehensive analysis of desired cell types in a tissue may require the use of a combination of complementary, individually optimized preparation methods and sequencing approaches as presented here for SECs. We utilized bulk- RNA-seq of purified SECs from C57BL/6 J mice as well as our current best optimized methods for scRNA-seq and snRNA-Seq to provide a robust analysis of the B6 limbus focusing on SEC transcriptomes. We successfully analyzed far greater numbers of SEC transcriptomes

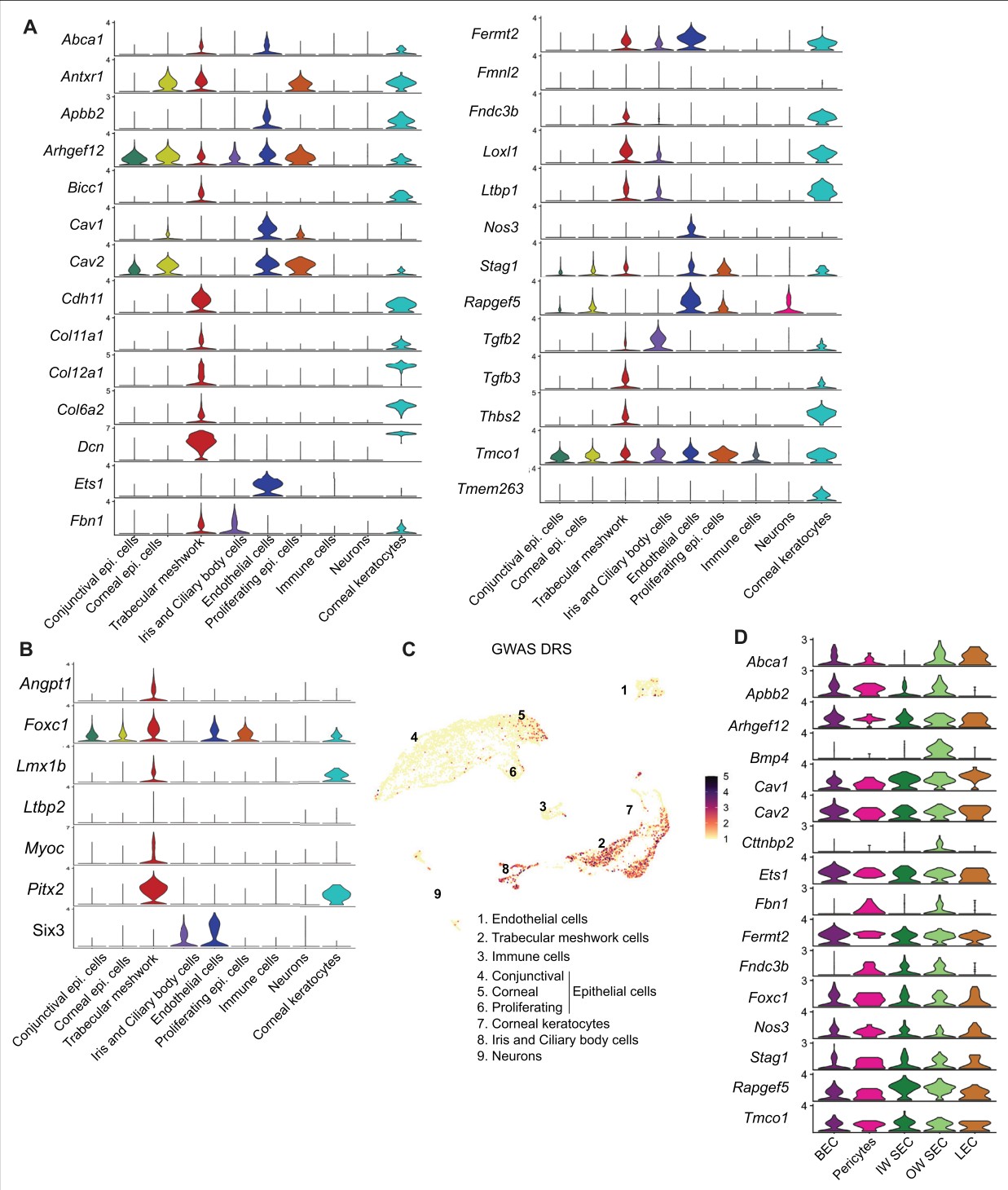

**Figure 6.** Cell-type assignment of high IOP and glaucoma genes. (**A**) Expression levels of gene list obtained from GWAS and genetic analyses associated with glaucoma and elevated IOP. (**B**) Expression levels of genes involved in developmental forms of glaucoma. (**C**) Disease relevance score (DRS) of genes from GWAS studies associated with glaucoma and elevated IOP. (**D**) Expression levels of specific genes in subtypes of endothelial cells.

The online version of this article includes the following figure supplement(s) for figure 6:

**Figure supplement 1.** Cell type assignment of high IOP and glaucoma genes.

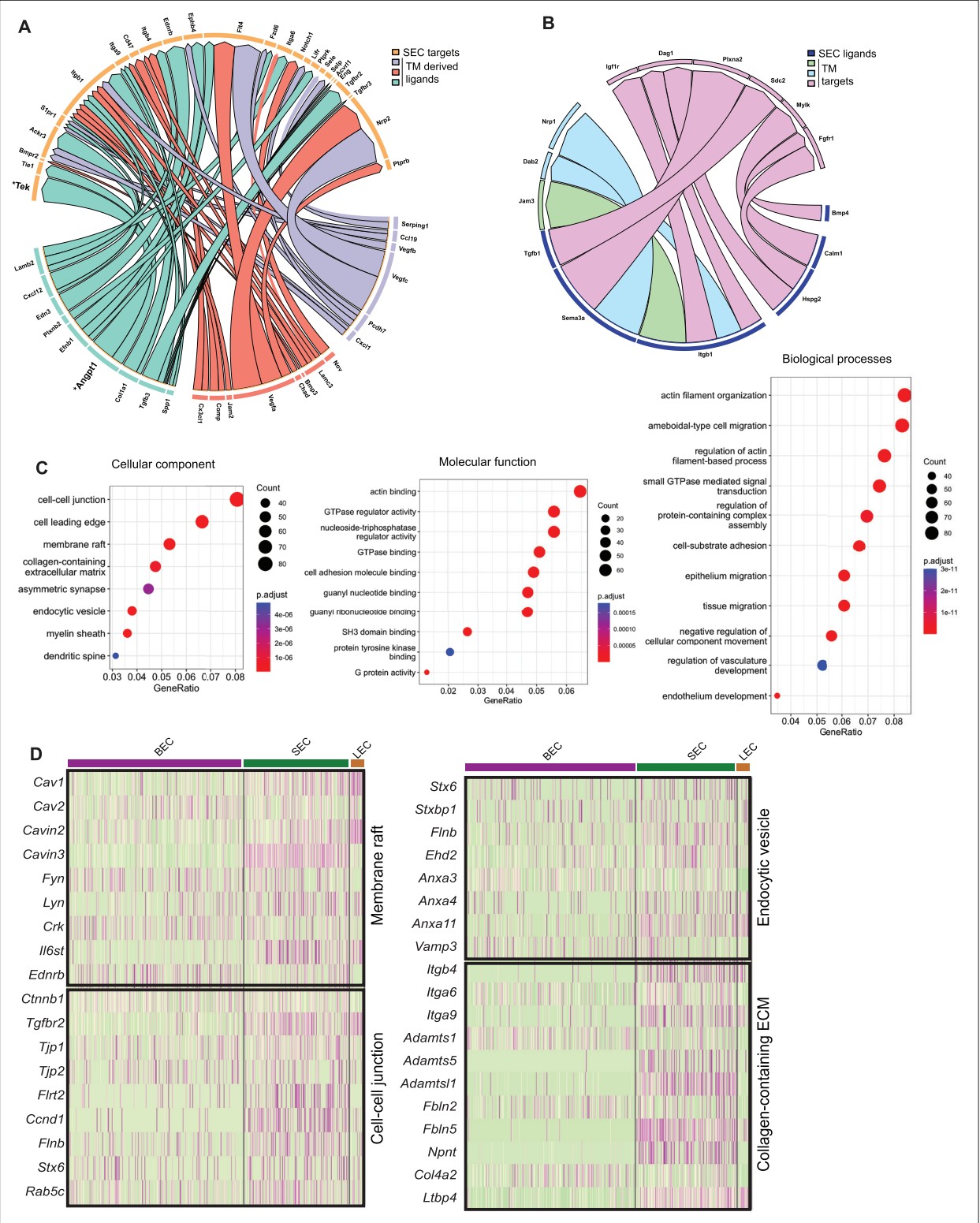

**Figure 7.** Predicted ligand-target analysis reveals signaling pairs between TM and SC. (**A**) (**B**) Circos plots depicting predicted ligand-target interaction between genes expressed in SECs and those in surrounding trabecular meshwork cells. (**C**) Gene ontology analysis of genes enriched in SC. (**D**) Heatmap of genes in individual enriched pathways showing their expression in endothelial cell subsets.

The online version of this article includes the following figure supplement(s) for figure 7:

**Figure supplement 1.** Gene ontology enrichment analysis highlights key cellular component/ pathways in SECs.

**Figure supplement 2.** Gene ontology enrichment analysis highlights key pathways in IW and OW SECs.

(541 total B6 transcriptomes, *Supplementary file 4* summarizes all datasets) than previous studies (*van Zyl et al., 2020*; *van Zyl et al., 2022*; *Patel et al., 2020*; *Thomson et al., 2021*).

## Deep transcriptome data and lymphatic biased identity of SECs

Despite their central role in assessing cellular heterogeneity, both snRNA-seq and scRNA-seq currently sample a much smaller proportion of the transcriptome than the averaged transcriptome provided by bulk-RNA-seq of a purified cell type. Our bulk-RNA-seq data provide the deepest analysis of the SEC transcriptome to date identifying >9000 expressed genes (for 3 biological replicates representing approximately 1500 cells each) at FPKM >5. Comparing this bulk SEC data to bulk data we generated for LECs and BECs (from the same limbal tissue samples) we observe a lymphatic biased identity for SECs. This differs from the conclusion in a recent scRNA-seq study (*Thomson et al., 2021*), but is consistent with previous studies showing expression of lymphatic regulatory molecules including *Prox1* and *Flt4* in SECs (*Kizhatil et al., 2014*; *Park et al., 2014*; *Aspelund et al., 2014*) and enrichment of lymphatic molecules in the IW of SC (*Kizhatil et al., 2014*). Adding to this, pseudo-bulk comparisons show that our scRNA-seq data agree well with our bulk data, again revealing the lymphatic biased expression pattern of SECs, and independently validating our conclusions based on bulk-seq. Our C57BL/6 J scRNA-seq is of high quality with greater transcriptome representation (median ≈2600 genes per cell) than is often achieved in limbal cells. Our re-analysis of the scRNA-seq data in the conflicting publication indicated that that study was underpowered for example capturing ~5 LECs and ~90 SECs (*Thomson et al., 2021*) and upon integration and analysis their SECs co-clustered with ours not shown, we did not include their cells in following studies as they differed from ours in being both from an albino strain and treated with Y27362, a ROCK inhibitor. Overall, our findings significantly expand knowledge of the molecular relationship between SECs, LECs and BECs (*Kizhatil et al., 2014*; *Stamer et al., 2015*) and they will support the field in formulating ideas and testing functions.

## snRNA-Seq allows identification of inner and outer wall SECs

snRNA-seq can have various advantages. It can allow the isolation of nuclei from frozen tissues. This theoretically helps to maintain transcriptomes as close to their natural state as possible and allows easy preservation of tissues for future analyses. Additionally, it can proportionally enrich for endothelial cells. For this reason, we used snRNA-seq in our studies.

ECs are very sensitive to preparation method and are routinely underrepresented using standard scRNA-seq methodologies, including in previous studies of the anterior segment and ocular drainage tissues (*van Zyl et al., 2020*; *Thomson et al., 2021*). On the other hand, snRNA-seq assays nuclear RNA, versus whole-cell RNA, and typically provides lower depth of transcriptome coverage than scRNA-seq. Our snRNA-seq data (strain C57BL/6 J) agree well with our bulk and scRNA-seq transcriptome data and again supports the lymphatic biased expression pattern of SECs. Our snRNA-seq data integrated well with the scRNA-seq data and provided proportionally far more SECs (Results). The added power of increased SEC numbers allowed us to distinguish the transcriptomes of IW and OW SECs for the first time.

## Enhanced identification of different SEC states using further scRNA-seq

In addition to sampling greater number of cells, deeper sampling of the transcriptome of each cell leads to more robust pathway analyses and improved resolution of heterogeneous cell states. Currently, popular data integration methods anchor datasets based on the genes that are sampled. This means that extending data by adding more cells with low sampling is not as informative as adding more cells with more deeply sampled transcriptomes (adding data with lower sampling may even be detrimental, with the balance between the added statistical power of increasing cell number and depth of transcriptome sampling being important). Despite the general value of snRNA-seq, it generally achieves lower depth of transcriptome coverage, with greater technical variation in sampling of specific genes. This means that snRNA-seq data generally provide lower power for pathway enrichment analyses than scRNA-seq (as is true in our data).

Thus, to both provide further scRNA-seq data to enhance the resolution of SEC subtypes and to allow a cross-strain comparison, we next sequenced limbal tissue from strain 129/Sj mice (*Tolman et al., 2021*). We obtained high-quality data with standard coverage (362 SECs and median 1144 genes

per cell, *Supplementary file 4*). The strain 129/Sj and B6 data integrate well and allowed stratification of SECs into three different states, including two IW states, for the first time. We named these classes OW, IW1, and IW2 based on marker studies and IF (see below). We refer to these SEC classes as states rather than subtypes because, despite their unbiased statistical separation, they have remarkably similar transcriptomes with no or very few detected pathway differences (at the currently achieved depth of sampling). Their transcriptomes may differ more due to the effects of local environmental differences (including exposure to different AQH flow parameters) than differing developmental trajectories. With greater sampling and deeper transcriptomic depth, it is likely that additional SEC cell states/types will be identified. Pathway enrichment analysis identified a few significantly different pathways between IW and OW cells. IW1 cells are enriched for the calcium-dependent protein binding and cell adhesion molecule binding pathways which makes intuitive sense, since SC contains many vesicles, suggesting that membrane fusion events must be important for normal function, likely in pore formation within drainage structures known as giant vacuoles. Membrane fusion in other cell types is regulated by calcium binding to various proteins including calmodulin (*Calm1*) (*Di Giovanni et al., 2010*). Inner wall cells have a discontinuous basement membrane and effective cell adhesion is necessary as the pressure gradient pushes them away from their basement membrane. Additionally, various cell adhesion molecule and related signaling can impact the stability of cell junctions, again with the possibility of modulating resistance to AQH drainage through the IW (*Kizhatil et al., 2023*).

## Marker genes for OW and IW cells

Analysis of our scRNA-seq and IF data identified *Selp* as an OW marker among SECs. In agreement with this, Weibel-Palade bodies that produce, store, and release P-Selectin (SELP) are enriched in OW SECs (*Hamanaka et al., 1992*). *Selp* was also expressed in CCs and some vascular endothelial cells. CCs also express *Ackr1* (*van Zyl et al., 2020*) and Results and have higher *Flt1* expression than OW and IW SECs. *Flt1* has been previously shown in SC (*Thomson et al., 2021*), with robust expression based on IHC in one study (*Sano et al., 2012*). *Flt1* RNA is expressed at low levels and its protein is undetectable by IF in most SECs in our study in both the strain C57BL/6 J and 129/Sj SECs. The reasons for this difference are not clear. Our integration and reanalysis of the scRNA-seq data from one study (*Thomson et al., 2021*) showed that their SECs had similar *Flt1* expression patterns to ours (not shown).

Notably no markers have completely unique gene expression in any of these SEC classes and a combination of markers will provide best specificity for the majority of SECs (*Figure 4E*). Despite this, and due to differences in expression levels, IF looking for robust expression is a reasonable initial approach using some of the key marker genes for which reliable antibodies are available (e.g. NPNT and CCL21A). Along these lines, our results agree with previous studies (*Thomson et al., 2021*) identifying *Npnt* as a marker for SECs among EC subtypes. *Npnt* promotes endothelial migration and tube formation (*Kuek et al., 2016*) and thus could be key in the formation and maintenance of the adult SC. We extend previous understanding, by showing that *Npnt* is enriched in both IW1 and IW2 in comparison to OW cells. In fact, *Npnt* is most highly expressed in IW1 cells, and its transcript was detected in 89% of IW1 versus 26% of IW2 and 32% of OW cells (*Figure 4E*, *Supplementary file 5*). *Ccl21a* was also recently reported in SECs (*Aspelund et al., 2014*; *van Zyl et al., 2020*). We extend this by showing that the *Ccl21a* gene is robustly expressed in IW2 cells but is typically expressed at low levels in IW1 and OW cells. Its transcript was detected in 45% of IW2 cells, 33% of IW1 and 21% of OW cells. Thus, high *Npnt* or high *Ccl21a* expression are key markers for these SEC classes (NPNT for IW1 and CCL21A for IW2). In agreement with our previous study demonstrating polarization of SC with a bias for robust expression of lymphatic molecules in the IW versus the OW (*Kizhatil et al., 2014*). IW cells have higher expression of key lymphatic genes including *Flt4*, *Prox1* and *Ccl21a*. Assessing various marker genes by IF confirmed the identification of these OW and IW cell states. Additionally, an analysis of SECs in publicly available dataset (*Thomson et al., 2021*), showed a similar distribution of *Npnt, Ccl21a,* and *Selp* expression in SECs (*Figure 5—figure supplement 1*) which further bolsters our findings.

## Potential regulators of AQH flow and immune cell transit expressed by IW SECs

In addition to validating the identification of OW and IW transcriptomes, our IF analyses further strengthened the biological separation of IW1 and IW2 SECs. IF revealed a bias for IW1 cells (marked

by robust NPNT) to be located more anteriorly in SC than IW2 cells (marked by robust CCL21A). IW2 cells have a bias to be localized more posteriorly in SC. In mice, the anterior portion of the SC IW is likely to experience higher transmural AQH flow rates based on the more open, anterior angle tissue anatomy, while the posterior portion of the IW is likely to experience lower flow rates. The posterior portion of SC has a differing local environmental context compared to the anterior portion being more closely opposed to: (1) the iris/uvea, (2) the more robust posterior trabecular meshwork, and (3) the ciliary muscle and ciliary body. Together, this suggests that *Npnt* may respond to (and/or be a marker of) either local AQH flow rates, to local transport of ligands in AQH that are carried by flow, or to other local environmental differences. Since it is an extracellular matrix molecule (*Toraskar et al., 2018*), NPNT may even directly modulate the local regulation of flow. Further studies are needed to assess these possibilities and are already underway. Interestingly, the IW2 marker CCL21A is a potent chemoattractant ligand. It engages the CCR7 receptor on lymphocytes and antigen-presenting cells (APCs). *Ccl21a* is prominently expressed in lymphatics and lymph nodes and directs the migration of immune cells into lymphoid tissues (*Russo et al., 2016*; *Tal et al., 2011*) Low fluid flow environments create a gradient of CCL21A that then directs cellular homing (*Russo et al., 2016*). We speculate that CCL21A may have similar functions in SC, which in addition to draining aqueous humor is intimately associated with immune cells, some of which transit the IW when exiting the eye (*Kizhatil et al., 2014*; *Cone and Pais, 2009*). SC is likely important in the transit of immune-tolerizing cells that are exposed to antigens in the anterior chamber, exit the eye and then travel to the spleen to establish a form of immune tolerance to the antigen captured in the eye, a process known as anterior chamber associated immune deviation (ACAID; *Cone and Pais, 2009*).

## Macrophages, pericytes, and collector channels

We previously reported a chaperoning role for macrophages in SC development, while adult SC is intimately associated with macrophages (*Kizhatil et al., 2014*). It seems likely that macrophages associated with SC and/or present in the TM will have important roles in determining AQH outflow/IOP in health and disease. Our scRNA-seq data did not have adequate power for pathway analysis of macrophages, but further studies are underway. Our scRNA-seq also identified pericytes. IF demonstrated that *Desmin*-expressing pericytes are present along the CCs. The physical association of pericytes with CCs suggests an important role in regulating outflow resistance at a site immediately distal to SC, but further experiments are needed to test this.

## Pathways in SECS

Gene ontology (GO) analysis of our scRNA-seq identified molecular pathways and hub genes that can be used to test hypotheses regarding SC function in regulating IOP. We identified genes involved in a variety of biological processes that are enriched in SECs. These processes include actin filament organization, cell substrate adhesion, regulation of vascular development, and tissue migration. Importantly, cell-cell junction pathway genes were enriched in SECs, including *Tjp1*, *Tjp2*, and *Ctnnb1*. These genes have important roles in the formation and regulation of tight junction proteins in other systems (*Richards et al., 2022*). Integrin genes involved in regulating collagen-containing extracellular matrix such as *Itga6*, *Itga9*, and *Itgb4*, fibulin genes with roles in maintaining the integrity of basement membrane such as *Fbln2* and *Fbln5*, and nephronectin (*Npnt*) an ECM molecule that regulates cell-cell adhesion (*Linton et al., 2007*) are all expressed in SECs. Our GO analysis indicates enrichment of 'membrane raft' pathway genes in SECs including *Fyn*, *Lyn*, and *Crk* kinases in SECs. Recently we demonstrated a role for FYN, a Src family kinase in regulating AQH outflow and IOP (*Kizhatil et al., 2023*). FYN is activated by IOP elevation and phosphorylates VECAD at specific tyrosine residues, thereby increasing the permeability of SEC cell junctions. Functionally analyzing hub genes identified in our pathway analyses is likely to improve our understanding of additional mechanisms that are active within SECs. Enhancing knowledge of pathway details in SECs, our bulk data provide much greater depth of transcript sampling and is valuable for extending the number of identified pathway members beyond the scRNA-seq data.

To provide important information on predicted ligand receptor signaling between the TM and SC, we used NicheNet (*Browaeys et al., 2020*) to identify matching ligand receptor pairs present on SECs and TM cells. A list of key ligand receptor pairs is presented in *Figure 7*. Supporting this approach and consistent with previous studies (*Kizhatil et al., 2014*; *Thomson et al., 2021*; *Thomson et al., 2017*),

this analysis shows that *Angpt1* is produced by TM cells while its receptor TEK is highly expressed in SECs. Disturbances in this pathway are known to result in IOP elevation and developmental glaucoma while GWAS implicates it in POAG. Since IOP elevation is a key risk factor for glaucoma, we also provide a summary of the key cell types in which key genes implicated in IOP elevation and glaucoma though GWAS are expressed.

In summary, our study provides a deep characterization of SECs including the identification of 3 distinct cell states of great value to understanding both SC biology and IOP elevation in glaucoma. Although we focus on SC here, a wealth of data is provided for other anterior segment cell types that will be the focus of future studies.

# Materials and methods

## Key resources table

| Reagent type (species) or resource | Designation | Source or reference | Identifiers | Additional information |
|---|---|---|---|---|
| Strain, strain background (*M. musculus*, both sexes) | 129/Sj | Internal | Ref PMID:33462143 | |
| Strain, strain background (*M. musculus*, both sexes) | C57BL/6 J | Jackson Labs | IMSR_JAX: 000664 | |
| Strain, strain background (*M. musculus*, both sexes) | Prox1-GFP BAC transgenic (Tg(Prox1-EGFP)KY221Gsat/Mmcd) | MMRRC | 031006-UCD | |
| Antibody | α-SMA (Rabbit polyclonal) | Abcam | Cat# ab5694; RRID:AB_2223021 | WM (1:50) Sections (1:200) |
| Antibody | CCL21A (Goat polyclonal) | R&D Systems | Cat# AF457-SP; RRID:AB_2072083 | WM (1:25) Sections (1:200) |
| Antibody | DARC/ACKR1 (Sheep polyclonal) | Invitrogen (Thermo Fisher Scientific) | Cat# PA5-47861; RRID:AB_2576815 | WM (1:50) Sections (1:200) |
| Antibody | Desmin (Rabbit monoclonal) | Invitrogen (Thermo Fisher Scientific) | Cat# MA532068; RRID:AB_2809362 | WM (1:50) Sections (1:200) |
| Antibody | Endomucin (Rat monoclonal) | EBioscience/Invitrogen (Thermo Fisher Scientific) | Cat# 14-5851-82; RRID:AB_891527 | WM (1:25) Sections (1:200) |
| Antibody | FLT1 (Rabbit monoclonal) | Abcam | Cat# ab32152; RRID:AB_778798 | WM (1:50) Sections (1:100) |
| Antibody | FLT4 (Goat polyclonal) | R&D Systems | Cat# AF743; RRID:AB_355563 | WM (1:50) Sections (1:200) |
| Antibody | LYVE1 (Rat monoclonal) | EBioscience/Invitrogen (Thermo Fisher Scientific) | Cat# 14-0443-80; RRID:AB_1633416 | WM (1:50) Sections (1:200) |
| Antibody | NPNT (Goat polyclonal) | R&D Systems | Cat# AF4298; RRID:AB_10645643 | WM (1:50) Sections (1:200) |
| Antibody | PECAM-1 (CD31) (Rat monoclonal) | BD Pharmingen | Cat# 550274; RRID:AB_393571 | WM (1:50) Sections (1:200) |
| Antibody | SELP (Goat polyclonal) | R&D Systems | Cat# AF737-SP, RRID: AB_2285644 | WM (1:50) Sections (1:100) |
| Antibody | VECAD (Goat polyclonal) | R&D Systems | Cat# AF1002; RRID:AB_2285644 | WM (1:50) Sections (1:200) |
| Commercial assay or kit | Papain Dissociation System | Worthington Biochemical | Cat# LK003153 | |
| Chemical compound, drug | 4% PFA | Thermo Fisher Scientific | Cat# 50-980-495 | |
| Chemical compound, drug | Propidium iodide | Thermo Fisher Scientific | Cat# P1304MP | |
| Chemical compound, drug | SYTOX green Nucleic Acid Stain | Thermo Fisher Scientific | Cat# S7020 | |
| Chemical compound, drug | Collagenase Type 4 | Worthington Biochemical | Cat# LS004188 | 1 mg/ml |
| Other | DAPI | Thermo Fisher Scientific | Cat# 62248 | (1:5000) Section 'Immunofluorescence' |
| Other | 40 and 100 µm Falcon cell strainers | Thermo Fisher Scientific | 08-771–1/08-771-19 | Section 'Processing of tissue for single nucleus RNA sequencing' |

## Mice

Prox1-GFP BAC transgenic mice (Tg(Prox1-EGFP)KY221Gsat/Mmcd) were maintained on C57BL/6 J (B6) background. B6 mice were obtained from JAX (Strain #:000664). The 129/Sj strain is maintained at our lab (see Key Resources Table; *Tolman et al., 2021*). Animal procedures were performed in accordance with the ARRIVE guidelines.

## Bulk RNA sequencing

The anterior segment of the eye including the limbus was dissected from each of ten 3-month-old Prox1-GFP mice to make one biological replicate of each cell type. The iris, cornea, and ciliary body were gently and carefully removed using forceps to prevent damage to the delicate aqueous humor drainage structures. A strip of limbal tissue 1–2 mm wide consisting of the aqueous humor outflow tissues SC and TM, limbal blood vessels, and lymphatics was dissected from each anterior segment. Limbal strips were dissected in ice-cold PBS. Strips from 20 eyes were pooled and dissociated paying strict attention to osmolarity (290–300 mOSm) and mechanical force, using 1 mg/ml Collagenase Type 4 (Worthington Biochemical Cat. no. LS004188) to preserve membrane proteins for the staining cell surface marker for flow cytometry. Single-cell suspension was stained with Lyve1 conjugated to eFluor 660 (ALY7 eBioscience Cat. no. 14-0443-82), Endomucin conjugated to phycoerythrin (eBio V.7.C7 Cat. no.14-5851-82), and propidium iodide to mark dead cells. Forward and side scatter gates were first used to eliminate events with low scatter which include derbies, cell fragments and pyknotic cells. Then propidium iodide positive dead cells were gated out. Further gating on the viable cells was applied such that distinct population of cells were isolated (a) SECs: $GFP^+Lvye1^-$, (b) LECs: $GFP^+$ $Lvye1^+$, (c) $GFP^-$ BECs: $Endomucin^+$. A total of 1500–1700 cells for SEC, 200–300 cells for LEC and 2000–2500 cells for BEC were collected. Three biological replicates were processed for each of SECs, BECs, and LECs. Raw data is available in GEO with accession number GSE272434.

## Processing of tissue for single-cell RNA sequencing

The anterior segment of the eye including the limbus was dissected from 3-month-old Prox1-GFP mice. A strip of limbal tissue 1–2 mm wide consisting of the aqueous humor outflow tissues SC and TM, ciliary body, iris, and cornea was dissected from each anterior segment in ice-cold Dulbecco's modified Eagle's medium DMEM. To enrich for aqueous humor outflow tissues, limbal strips from 6 C57Bl/6 J eyes (four females, two males) were pooled for a single run of C57BL/6 J scRNA-seq. C57BL/6 J scRNA-seq was performed on two separate pools of eyes for a total of 12 eyes (8 females, 4 males). For 129/Sj scRNA-seq, 6 eyes (3 females, 3 males) were pooled for a single run.

Single cell dissociation was performed using Papain and Deoxyribonuclease I (Worthington Biochemical Cat. no. LK003153) for 20 min at 37 °C and stopped using Earl's balanced salt solution (EBSS- Thermo Fisher Scientific, Cat no. 24010–043). Cells were triturated using an 18-gauge needle, centrifuged at 300 × *g* at 4 °C, washed with cold DMEM, and filtered using a 100 µm cell strainer. Cells were resuspended in cold DMEM and placed on ice immediately. Cells were counted using Countess II automated cell counter and processed for 10 x library preparation immediately (see below).

## Processing of tissue for single nucleus RNA sequencing

The anterior segment of the eye including the limbus was dissected from 3-month-old Prox1-GFP mice. A strip of limbal tissue 1–2 mm wide consisting of the aqueous humor outflow tissues SC and TM, ciliary body, iris, and cornea was dissected from each anterior segment in ice-cold Dulbecco's modified Eagle's medium DMEM. To enrich for aqueous humor outflow tissues, limbal strips from 6 eyes (4 females, 2 males) were pooled for a single run of snRNA sequencing. Dissociation was performed using Papain and Deoxyribonuclease I (Worthington Biochemical Cat. no. LK003153) for 20 min at 37 °C and stopped using Earl's balanced salt solution (EBSS). Cells were placed on ice and further nuclei lysis steps were performed according to 10 X nuclei isolation from cell suspensions protocol (CG000124). Briefly, cells were lysed in lysis buffer for 45 s, resuspended in nuclei wash and resuspension buffer, and centrifuged for 10 min at 500rcf at 4 °C. Pellet was resuspended in nuclei wash and resuspension buffer and passed through a 40 µm cell strainer. An aliquot of nuclei suspension was stained with SYTOX green to count nuclei using Countess II automated cell counter.

## Single-cell and single-nucleus RNA sequencing

Single-cell and single-nucleus RNA-seq was performed at the single-cell sequencing core in Columbia Genome Center. Single cells or nuclei were loaded into chromium microfluidic chips with v3 chemistry and barcoded with a 10 x chromium controller (10 x Genomics). RNA from the barcoded cells was reverse-transcribed, and sequencing libraries were constructed with a Chromium Single Cell v3 reagent kit (10 x Genomics). Sequencing was performed on NovaSeq 6000 (Illumina). Raw data is available in GEO with accession number GSE271132.

## Analysis of sequencing data

### Bulk RNA sequencing

Raw reads were aligned to mm10 reference genome by STAR and summarized to gene counts. After transcripts Per Kilobase Million (TPM) normalization, principal components analysis (PCA) was applied to assess technical replication and remove outlier samples. DESeq2 models were fitted to identify genes differentially expressed between blood vessel endothelial, SC endothelial, and lymphatic endothelial cells, where a threshold of log (fold change)>2 and FDR <0.05 was used to identify a total of 3063 variable genes between three pair-wise comparisons. TPM of differentially expressed genes were shown as a heatmap between samples. The resulting log (fold change) values for each gene were plotted against those obtained from single-cell sequencing data described below.

### Single-cell and single-nucleus RNA sequencing

Raw reads were mapped to the mm10 reference genome by 10 x Genomics Cell Ranger pipeline. Seurat v3 (*Satija et al., 2015*; *Butler et al., 2018*; *Stuart et al., 2019*; *Hao et al., 2021*) was used to conduct all single-cell and single-nucleus sequencing analyses. Briefly, the dataset was filtered to contain cells with at least 200 expressed genes and genes with expression in more than three cells. Cells were also filtered for mitochondrial gene expression (<20% for single-cell and <5% for single-nucleus, mitochondrial genes are rarely represented in snRNA-seq). The dataset was log-normalized and scaled. Unsupervised clustering was performed using a resolution parameter of 0.1, followed by manual annotation of Seurat clusters. Sub-clustering of individual clusters was also performed in an unsupervised manner with a resolution parameter of 0.5. Biological incompatibility based on gene expression was used to identify doublets. The two C57BL/6 J single-cell datasets were merged after QC. They were technical replicates, experimentally processed at the same time, and so batch effects resulting from sample preparation, sequencing technology, etc. do not apply. The merged C57BL/6 J singlecell dataset was integrated with the C57BL/6 J single nucleus dataset post-QC, because of expected batch effects between datasets arising from differences in sample preparation, sequencing technology, and alignment of reads (inclusion of intronic reads in snRNA-seq). Similarly, the merged C57BL/6 J single cell dataset was integrated with the 129/Sj single-cell dataset post-QC. Briefly, samples were log-normalized followed by the selection of highly variable genes and a reduction of dimensions using canonical correlation analysis. The unsupervised clustering with a resolution parameter of 0.1 for both single-cell and single-nucleus sequencing data was represented on a common UMAP space, and cluster identity was assigned based on the expression of various known genes. Marker genes were identified using Wilcoxon test implemented in Seurat using default parameters.

### scDRS

From genome-wide association studies of glaucoma (*Khawaja et al., 2018*; *MacGregor et al., 2018*; *Choquet et al., 2018*; *Gao et al., 2018*; *Gharahkhani et al., 2021*; *Shiga et al., 2018*) we curated a list of genes associated with the disease and applied single-cell disease relevance score (*Zhang et al., 2022*) to assess glaucoma disease risk for each single cell, using default parameters.

### GO analysis

Gene ontology and pathway enrichment analysis was performed by generating a list of genes enriched in individual endothelial cell groups by comparing against a background expression 'universe' of genes expressed in endothelial cells and trabecular meshwork cells (p value cut-off 0.01) (*Timmons et al., 2015*) using R package ClusterProfiler (*Yu et al., 2012*).

### GSEA analysis

Gene set enrichment analysis (GSEA) was performed with a pre-ranked list of genes from bulk RNA sequencing data with adjusted p-values ≤0.05 and log fold change between groups ≥2 were considered significantly different. GSEA was performed against canonical pathway consisting of 3090 gene sets.

### Predicted ligand-receptor interactions

Predicted ligand-target links between interacting cells was performed using NicheNet (*Browaeys et al., 2020*). Briefly, expression of genes in cell types is linked to a database on signaling and gene regulatory networks curated based on prior information to make viable predictions on potential interactions between cell types.

## Immunofluorescence

Whole mounts of anterior segment of the eye were obtained from B6. Prox1-GFP mice or B6 mice and processed as described in *Kizhatil et al., 2014*. Briefly, mice were cervically dislocated and eyes harvested with the optic nerve intact and fixed in ice-cold 4% PFA for 2 hr at 4 °C. The anterior segment including the limbus was dissected in ice-cold PBS. The iris was carefully removed and shallow cuts were made to the anterior eye cup to display four quadrants in a petal-like pattern. The tissue was incubated in blocking solution of 0.3% PBST with 3% BSA for 2 hr at room temperature. Primary antibodies in blocking solution were added and the eyes left at 4 °C for 2–3 days. After four PBST washes of an hour each at room temperature, secondary antibodies along with DAPI were added and eyes placed in the dark at 4 °C overnight. After four PBS washes of an hour each at room temperature, eyes were mounted on a slide, cover-slipped, and imaged.

For sections, B6. Prox1-GFP mice or B6 mice were cervically dislocated and eyes harvested with the optic nerve intact and fixed in ice-cold 4% PFA for 2 hr in 4 °C. A window was made to the back of the eye cup by removing the optic nerve. Eyes were equilibrated in 30% sucrose and embedded in OCT. Eyes were stored at –80 °C until they were sectioned and processed for immunostaining. Sections were briefly washed in PBS and 0.3% PBST and blocked with blocking buffer made of 10% donkey serum in PBST. Primary antibodies were applied in blocking buffer overnight at 4 °C and secondary antibodies with DAPI for 2 hr at room temperature. Slides were washed and cover-slipped.

## RNAscope in situ hybridization

In situ hybridization was carried out using RNAscope Multiplex Fluorescent Reagent Kit v2-Mm (Advanced Cell Diagnostics a Bio Techne Brand, Newark, CA). Probes for mouse NPNT and SELP (also purchased from ACD Bio) were used. Fluorescent dyes Opal 690 and Opal 620 were used with each of these probes. Briefly, eyes from 3-month-old C57BL/6 J were enucleated and fixed in 4% PFA for 24 hr at 4 °C. A window was made to the back of the eye cup, lens removed, and the anterior cup placed in 30% sucrose overnight at 4 °C. Tissue was cryopreserved in OCT and placed at –80 °C. Sections were cut at 12 µm thickness. RNAscope was performed over 2 days as per manufacturer's protocol. Z stack images were taken using a confocal microscope at ×40 magnification. A maximum projection image was then taken of the completed Z stack and edited using ImageJ.

## Confocal microscopy and postprocessing of images

Sections and whole mounts were imaged on a Leica SP8 confocal. Images were acquired at 40 X resolution. 30–45 um Z-stack images of the tissue were acquired from whole mounts to image from the vascular plexus down to the trabecular meshwork cells. Images were analyzed and processed in Imaris.

## Acknowledgements

We are grateful to Drs. Ross Ethier (Georgia Institute of Technology), and Darryl Overby (Imperial College London) for critical reading of the manuscript. The project was funded by BrightFocus Foundation grants BFOCUS CG2020004 (KK, SJ, DS) and NGR G2021007S (RB) with partial support from National Eye Institute (NEI) grants R01EY028175 (KK), R01EY032507 (SJ), R01EY032062 (KK,

SJ), R01EY029548 (JQ), R01EY022359 (DS), New York Fund for Innovation in Research and Scientific Talent (NYFIRST; EMPIRE CU19-2660) award (SJ), and startup funds at Columbia University, including the Precision Medicine Initiative. The content is solely the responsibility of the authors and does not necessarily represent the official views of the National Institutes of Health. Acknowledgment is made to the donors of the NGR, a program of the BrightFocus Foundation, for support of this research. The project was also supported by a Vision Core grant P30EY019007 (Columbia University), P30EY005722 (Duke University) and an unrestricted departmental award from Research to Prevent Blindness (Columbia University).

## Additional information

### Funding

| Funder | Grant reference number | Author |
| --- | --- | --- |
| BrightFocus Foundation | CG2020004 | Krishnakumar Kizhatil W Daniel Stamer Simon WM John |
| BrightFocus Foundation | G2021007S | Revathi Balasubramanian |
| National Eye Institute | R01EY028175 | Krishnakumar Kizhatil |
| National Eye Institute | R01EY032507 | Simon WM John |
| National Eye Institute | R01EY032062 | Krishnakumar Kizhatil Simon WM John |
| National Eye Institute | R01EY029548 | Jiang Qian |
| National Eye Institute | R01EY022359 | W Daniel Stamer |
| New York Fund for Innovation in Research and Scientific Talent | EMPIRE CU19-2660 | Simon WM John |

The funders had no role in study design, data collection and interpretation, or the decision to submit the work for publication.

### Author contributions

Revathi Balasubramanian, Conceptualization, Data curation, Formal analysis, Funding acquisition, Investigation, Methodology, Writing – original draft, Writing – review and editing; Krishnakumar Kizhatil, Conceptualization, Data curation, Funding acquisition, Methodology, Writing – review and editing; Taibo Li, Nicholas Tolman, Marina Simón, Formal analysis, Writing – review and editing; Aakriti Bhandari, Graham Clark, Violet Bupp-Chickering, Ruth A Kelly, Sally Zhou, John Peregrin, Formal analysis; Christa Montgomery, Writing - original draft, Writing – review and editing; W Daniel Stamer, Jiang Qian, Supervision, Funding acquisition, Writing – review and editing; Simon WM John, Conceptualization, Supervision, Funding acquisition, Writing – review and editing

### Author ORCIDs

Revathi Balasubramanian (ID) https://orcid.org/0000-0002-2209-0815
Taibo Li (ID) https://orcid.org/0000-0002-6624-9293
Marina Simón (ID) http://orcid.org/0000-0001-8133-8668
Christa Montgomery (ID) https://orcid.org/0009-0004-2430-2772
Simon WM John (ID) https://orcid.org/

### Ethics

This study was performed in strict accordance with the recommendations in the Guide for the Care and Use of Laboratory Animals of the National Institutes of Health. All animal procedures were performed according to the protocols approved by Columbia University's Institutional Animal Care and Use Committee (AABE9554 and AABU0654) and followed the ARRIVE guidelines.

Reviewer #1 (Public review): https://doi.org/10.7554/eLife.96459.3.sa1

Reviewer #2 (Public review): https://doi.org/10.7554/eLife.96459.3.sa2

Author response https://doi.org/10.7554/eLife.96459.3.sa3

# Additional files

## Supplementary files

- Supplementary file 1. GSEA of SECs compared to LECs.
- Supplementary file 2. GSEA of SECs compared to BECs.
- Supplementary file 3. DEG list comparing SECs of 129/Sj and C57BL/6 J datasets.
- Supplementary file 4. Comparison of SEC transcriptomes across studies.
- Supplementary file 5. DEG list comparing IW1 and IW2 SECs in C57BL/6 J.
- MDAR checklist

## Data availability

Sequencing data have been deposited in GEO under accession codes GSE272434 and GSE271132. Data can be visualized in the Broad Institute's Single Cell Portal with accession numbers SCP2293 (cell) and SCP2317 (nucleus). Publicly available data from previous studies (*Thomson et al., 2021*) were obtained from GEO under the accession code GSE168200.

The following datasets were generated:

| Author(s) | Year | Dataset title | Dataset URL | Database and Identifier |
|---|---|---|---|---|
| Revathi B, Krishnakumar K, Taibo L, Simon J | 2024 | Transcriptomic profiling of Schlemm's canal cells reveals a lymphatic-biased identity and three major cell states | https://www.ncbi.nlm.nih.gov/geo/query/acc.cgi?acc=GSE272434 | NCBI Gene Expression Omnibus, GSE272434 |
| Li T, Balasubramanian R, Kizhatil K, Qian J, John S | 2024 | Transcriptomic profiling of Schlemm's canal cells reveals a lymphatic-biased identity and three major cell states | https://www.ncbi.nlm.nih.gov/geo/query/acc.cgi?acc=GSE271132 | NCBI Gene Expression Omnibus, GSE271132 |

The following previously published dataset was used:

| Author(s) | Year | Dataset title | Dataset URL | Database and Identifier |
|---|---|---|---|---|
| Thomson BR, Liu P, Onay T, Du J, Tompson SW, Young TL, Jin J, Quaggin SE | 2021 | Cellular crosstalk regulates the aqueous humor outflow pathway and provides new targets for glaucoma therapies | https://www.ncbi.nlm.nih.gov/geo/query/acc.cgi?acc=GSE168200 | NCBI Gene Expression Omnibus, GSE168200 |

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
