## [Editor Report · eLife assessment]

This **valuable** study has characterized the unique expression of Schlemm's canal endothelial cells (SECs) using FACS-sorted specific cell bulk RNA-Seq and scRNA-/snRNA-Seq of mouse SECs. The **compelling** study identified novel biomarkers for SECs and molecular markers for two inner wall SEC states and outwall SECs in mouse eyes. Significant gene networks and pathways were elucidated for their potential contribution to glaucoma pathogenesis, providing targets for further research in relation to glaucoma.

---

## [Referee Report · Reviewer #1 (Public review)]

Summary:

Balasubramanian et al. characterized the cell types comprising mouse Schlemm's canal (SC) using bulk and single cell RNA sequencing (scRNA-seq). The results identify expression patterns the delineate the SC inner and outer wall cells and two inner wall 'states'. Further analysis demonstrates expression patterns of glaucoma associated genes and receptor ligand pairs between SEC's and neighboring trabecular meshwork.

Strengths:

While mouse SC has been profiled in previous scRNA-seq studies (van Zyl et al 2020, Thomson et al 2021), these data provide higher resolution of SC cell types, particularly endothelial cell (SEC) populations. SC is an important regulator of anterior chamber outflow and has important consequences for glaucoma.

Comments on the latest version:

The authors have addressed my primary concerns with the first version of the manuscript. This study represents a valuable resource in the molecular characterization of mouse Schlemm's canal cell types.

---

## [Referee Report · Reviewer #2 (Public review)]

Summary:

This revised article has characterized the mouse Schlemm's canal expression profile using a comprehensive approach based on sorted SEC, LEC, and BEC total RNA-Seq, scRNA-Seq, and snRNA-Seq to enrich the selection of SECs. The revised study has successfully profiled genome-wide gene expression using sorted SECs, demonstrating that SECs have a closer similarity to LECs than BECs. The combined scRNA- and snRNA-Seq data with deep coverage of gene expression led to the successful identification of many novel biomarkers for inner wall SECs, outer wall SECs, collector channel ECs, and pericytes. In addition, the study also identified two novel states of inner wall SECs separated by new markers. The study provides significant novel information about the biology and expression profile of SECs in the inner and outer walls. It is of great significance to have this novel, convincing, and comprehensive study led by leading researchers published in this journal. The revision has improved the clarity and significance of the study with more details.

Strengths:

This is a comprehensive study using various data to support the expression characterization of mouse SECs. First, the study profiled genome-wide expression using sorted SECs, LECs, and BECs from the same tissue/organ to identify the similarities and differences among the three types of cells. Second, snRNA-Seq was applied to enrich the number of SECs from mouse ocular tissues significantly. Increased sampling of SECs and other cells led to more comprehensive coverage and characterization of cells, including pericytes. Third, the combined scRNA- and snRNA-Seq data analyses increase the power to further characterize the subtle differences within SECs, leading to identifying the expression markers of Inner and Outer wall SECs, collector channel ECs, and distal region cells. Fourth, the identified unique markers were validated for RNA and protein expression in mouse ocular tissues. Fifth, the study explored how the IOP- and glaucoma-associated genes are expressed in the ScRNA- and snRNA-Seq data, providing potential connections of these GWAS genes with IOP and glaucoma. Sixth, the initial pathway and network analyses generated exciting hypotheses that could be tested in other independent studies.

Weaknesses:

The authors have addressed most of the previous comments by adding more details about the protocol and additional discussions. Several comments requiring additional experimental data have been addressed as future directions, such as protein validation, RNA expression validation in human samples, and GWAS-identified IOP genes.

Comments on the latest version:

The authors have addressed previous comments responsively. The authors have suggested several experiments to be completed in the future since these could be time-consuming with human samples. The revised article is with better clarity and clearer significance. No additional comments.

---

## [Author Response]

The following is the authors’ response to the original reviews.

**Reviewer #1 (Public Review):**
Summary:Balasubramanian et al. characterized the cell types comprising mouse Schlemm's canal (SC) using bulk and single-cell RNA sequencing (scRNA-seq). The results identify expression patterns that delineate the SC inner and outer wall cells and two inner wall 'states'. Further analysis demonstrates expression patterns of glaucoma-associated genes and receptor-ligand pairs between SEC's and neighboring trabecular meshwork.Strengths:While mouse SC has been profiled in previous scRNA-seq studies (van Zyl et al 2020, Thomson et al 2021), these data provide higher resolution of SC cell types, particularly endothelial cell (SEC) populations. SC is an important regulator of anterior chamber outflow and has important consequences for glaucoma.

We thank the reviewers for their thorough reading of our manuscript and their insightful comments.

Weaknesses:(1) Since SC has previously been characterized in mouse, human, and other species by scRNA-seq in other studies, this study would benefit from more direct comparisons to published datasets. For example, Table 4 could be expanded to list the SC cell numbers profiled in each study. Expression patterns highlighted in this study could be independently verified by plotting in publicly available mouse SC datasets. Further, a comparison to human expression patterns would assess whether type-specific expression patterns are conserved. Alternatively, an integrated analysis could be performed. Indeed, the authors mention that an integrated analysis was attempted but the data is not shown. It is unclear if this was because of a lack of agreement between datasets or other reasons.

Table 4 now includes an expanded list of SC cell numbers in each study. We profiled the expression of Npnt, Selp, and Ccl21a in the Thomson et al., 2021 dataset and have included the concurring results in Figure S5. We were unable to do a similar profile using the Van Zyl., 2020 dataset due to small SC numbers. As previously mentioned, differences such as read depth, strain of animals used (including pigmented vs albino), method of cell isolation (including drug exposure), and number of cells profiled raise a significant impediment to integration with previously published datasets. A comparison to human atlas is a focus of future work.

(2) Figure 1 presents bulk RNA seq results comparing SEC, BEC, and LEC expression patterns. These populations were isolated using cell surface markers and enrichment by FACS. Since each EC population is derived from the same sample, the accuracy of this data hinges on the purity of enrichment. However, a reference is not given for this method and it is not clear how purity was validated. The authors later note that marker Emcn, which was used to identify BECs, is also expressed in SECs and LECs at lower levels. It should be demonstrated that these populations are clearly separated by flow cytometry.

We have added the following clarifying text to the methods section: Forward and side scatter gates were first used to eliminate events with low scatter which include debris, cell fragments and pyknotic cells. Then propidium iodide positive dead cells were gated out. Further gating on the viable cells was applied such that distinct population of cells were isolated (a) SECs: GFP+Lvye1-, (b) LECs: GFP+ Lyve1+, (c) GFP- BECs: Endomucin+.

We show here a representative of the flow sort showing the clear distinction in SEC and LEC cell isolation.

**Author response image 1. sa3fig1:** Flow sorted SEC and LEC. We obtained two distinct populations; 1. SEC cells (GPF+LYVE1--blue) 2. LEC (GPF+LYVE1+- red). Note eFluor 660 emission was collected using the Alexa647 (A647) setting of the flow cytometer. Additionally, SEC marker expression from bulk RNA-seq aligns with signature gene expression from SECs in single cell RNA-seq (Figure S3).

(3) Bulk RNA-seq analysis infers similarity from the number of DEGs between samples, however, this is not a robust indicator. A correlation analysis should be run to verify conclusions.

We have provided a heatmap with hierarchical clustering based on Euclidean distance of the EC subtypes (Figure 1B) analyzed by bulk RNA seq in addition to number of DE genes between subtypes.

(4) Figures 2-4 present three different datasets targeting the same tissue: (1) C57bl/6j scRNA-seq, (2) C57bl/6j snRNA-seq, (3) 129/sj scRNA-seq. Integrated analysis comparing datasets #1 to #2 and #3 is also presented. Integration methods are not described beyond 'normalization for cell numbers'. It is unclear if additional alignment methods were used. Integration across each of these datasets needs careful consideration, especially since different filtering methods were used (e.g. <20% mito in scRNA-seq and <5% in snRNA-seq). Improper integration could affect the ability to cluster or exaggerate differences between cell/types and states. It would be useful to demonstrate the contribution of different samples and datasets to each cell type/state to verify that these are not driven by batch effects, mouse strain, or collection platform.

We agree that integration should be performed with careful consideration to confounding factors. We demonstrate the contribution of different samples and datasets to show how our datasets integrated well (we had added panels to Figure 3C and 4C) and that cell types/states contribution was uniformly distributed across methods (C57BL/6J single cell and single nuc) and backgrounds (C57BL/6J and 129/Sj) were not a result of integration.

(5) IW1 and IW2 are not well separated, and it is unclear if these represent truly different cell states. Figure 5b shows the staining of CCL21A and describes expression in the 'posterior portion' but in the image there are no DAPI+ nuclei in the anterior portion, suggesting the sampling in this section is different from Figure 5a. This would be improved by co-staining NPNT and CCL21A to demonstrate specificity.

Since both our antibodies are derived from the same species (goat), a co-labeling wasn’t possible. To be prudent, we used adjacent sections, flat-mounts, and RNAscope and provided further evidence of the anterior/posterior “bias” in supplemental figures.

(6) The substructures observed within clusters in sc/snRNA-seq data suggest that overall profiling may still not be comprehensive. This should be noted in the discussion.

We agree and have added this note in the discussion: “With greater sampling and deeper transcriptomic depth, it is likely that additional SEC cell states/types will be identified.”

**Reviewer #2 (Public Review):**
Summary:This article has characterized the mouse Schlemm's canal expression profile using a comprehensive approach based on sorted SEC, LEC, and BEC total RNA-Seq, scRNA-Seq, and snRNA-Seq to enrich the selection of SECs. The study has successfully profiled genome-wide gene expression using sorted SECs, demonstrating that SECs have a closer similarity to LECs than BECs. The combined scRNA- and snRNA-Seq data with deep coverage of gene expression led to the successful identification of many novel biomarkers for inner wall SECs, outer wall SECs, collector channel ECs, and pericytes. In addition, the study also identified two novel states of inner wall SECs separated by new markers. The study provides significant novel information about the biology and expression profile of SECs in the inner and outer walls. It is of great significance to have this novel, convincing, and comprehensive study led by leading researchers published in this journal.Strengths:This is a comprehensive study using various data to support the expression characterization of mouse SECs. First, the study profiled genome-wide expression using sorted SECs, LECs, and BECs from the same tissue/organ to identify the similarities and differences among the three types of cells. Second, snRNA-Seq was applied to enrich the number of SECs from mouse ocular tissues significantly. Increased sampling of SECs and other cells led to more comprehensive coverage and characterization of cells, including pericytes. Third, the combined scRNA- and snRNA-Seq data analyses increase the power to further characterize the subtle differences within SECs, leading to identifying the expression markers of Inner and Outer wall SECs, collector channel ECs, and distal region cells. Fourth, the identified unique markers were validated for RNA and protein expression in mouse ocular tissues. Fifth, the study explored how the IOP- and glaucoma-associated genes are expressed in the ScRNA- and snRNA-Seq data, providing potential connections of these GWAS genes with IOP and glaucoma. Sixth, the initial pathway and network analyses generated exciting hypotheses that could be tested in other independent studies.

We thank the reviewer for their comments on the strengths of this study.

Weaknesses:A few minor weaknesses have been noted. First, since snRNA-Seq and scRNA-Seq generated different coverage of expressed genes in the cells, how did the combined analyses balance the un-equal sequencing coverage and missing data points in the snRNA-Seq data? Second, the RNA/protein validation of selected SEC molecular markers was done using mouse anterior segment tissues. It would be more helpful to examine whether these molecular markers for SECs could work well in human SECs. Third, the effort to characterize the GWAS-identified IOP- and glaucoma-associated genes is exciting but with limited new information. Additional work could be performed to prioritize these genes.

Integration of sc-Seq and sn-Seq data: We have addressed a similar integration question from reviewer 1 and have now included a plot showing the distribution of cells upon integration. Integration methods are not perfect and generally result in some loss of data especially when datasets of un-equal sequencing coverage are integrated. However, we did not observe any obvious differences between the original (un-integrated) and integrated datasets. We also noted that cell types/states contribution was similarly distributed across methods (C57BL/6J single cell and single nuc) and backgrounds (C57BL/6J and 129/Sj) and clustering were not a result of batch-effects.

We agree about the human relevance of SEC markers, and this will be a focus of future work.

Another focus of our future work is to understand how GWAS identified IOP and glaucoma genes change in disease states.

**Recommendations for the authors:**

**Reviewer #1 (Recommendations For The Authors):**
Minor:(1) Figure 5- DAPI should be listed in the legend.(2) Figure 5- It would be helpful to label IW1 and IW2 regions in the UMAPs.

We have incorporated the suggestions in Figure 5 and legend.

**Reviewer #2 (Recommendations For The Authors):**
(1) The study has validated RNA/protein expression of the selected biomarkers for IW/OW SECs in mouse eyes. It would be more helpful to confirm that these newly identified molecular biomarkers for SECs could apply to human eyes. This could be examined through available human scRNA-/snRNA-Seq data or targeted RNA and protein staining experiments. The additional validation in human SECs would make the current discovery more convincing.

We agree with the importance of validation in human samples, and is the scope of future work.

(2) The combination of scRNA-Seq and snRNA-Seq from three batches of experiments increased the statistical power to identify subtypes of SECs. It would be helpful to include more details on how the qc, missing data, and normalization across different batches were dealt with.

We have incorporated more details in the methods section of the paper.

(3) The authors explored the underlying molecular connection between the newly identified IOP/glaucoma-associated genes using the newly generated SEC-targeted scRNA/snRNA-Seq data. Many of these associated genes were present in the same SEC cells. It would be interesting to see how many of these genes' expression levels are correlated with each other via a network. These potential correlation networks across SECs could lead to identifying novel upstream regulators or network hubs, which could target many IOP-associated genes for future studies.

We agree with the importance of a correlation network analysis, but this is a focus of future work, especially in normal and disease states.